# Noise Hypernetworks: Amortizing Test-Time Compute in Diffusion Models

**Luca Eyring**[1,2,3]    **Shyamgopal Karthik**[1,2,3,4]    **Alexey Dosovitskiy**[5]
**Nataniel Ruiz**[6*]    **Zeynep Akata**[1,2,3*]

[1]Technical University of Munich    [2]Munich Center of Machine Learning
[3]Helmholtz Munich    [4]University of Tübingen    [5]Inceptive    [6]Google
`luca.eyring@tum.de`

## Abstract

The new paradigm of test-time scaling has yielded remarkable breakthroughs in Large Language Models (LLMs) (e.g. reasoning models) and in generative vision models, allowing models to allocate additional computation during inference to effectively tackle increasingly complex problems. Despite the improvements of this approach, an important limitation emerges: the substantial increase in computation time makes the process slow and impractical for many applications. Given the success of this paradigm and its growing usage, we seek to preserve its benefits while eschewing the inference overhead. In this work we propose one solution to the critical problem of integrating test-time scaling knowledge into a model during post-training. Specifically, we replace reward guided test-time noise optimization in diffusion models with a Noise Hypernetwork that modulates initial input noise. We propose a theoretically grounded framework for learning this reward-tilted distribution for distilled generators, through a tractable noise-space objective that maintains fidelity to the base model while optimizing for desired characteristics. We show that our approach recovers a substantial portion of the quality gains from explicit test-time optimization at a fraction of the computational cost. Code is available at https://github.com/ExplainableML/HyperNoise.

## 1   Introduction

Recently, inference-time scaling has made remarkable breakthroughs in Large Language Models [28, 40, 84] and generative vision models, enabling models to spend more computation during inference to solve complex problems effectively. Drawing from the success and growing usage of test-time compute in LLMs, several methods have attempted to apply similar ideas in the context of diffusion models for generation [6, 22, 59, 66, 67, 80, 88, 90, 92, 93, 97]. The goal of this process is to spend additional compute during inference to obtain generations that better reflect desired output properties.

Diffusion model test-time techniques that optimize the initial noise or intermediate steps of the diffusion process, often guided by feedback from pre-trained reward models [48, 54, 101, 102, 103, 107], have demonstrated significant promise in improving critical attributes of the generated outputs, such as prompt following, aesthetics, quality and composition [11, 22, 44, 59, 66, 67, 97]. These methods generally fall into two broad categories: gradient-based optimization, which typically requires substantial GPU memory for backpropagation through the full model [6, 22, 46, 67, 97], and gradient-free optimization, which often necessitates a very large number of function evaluations (NFEs), sometimes thousands, of the computationally expensive denoising network [44,

---

*Equal Supervision

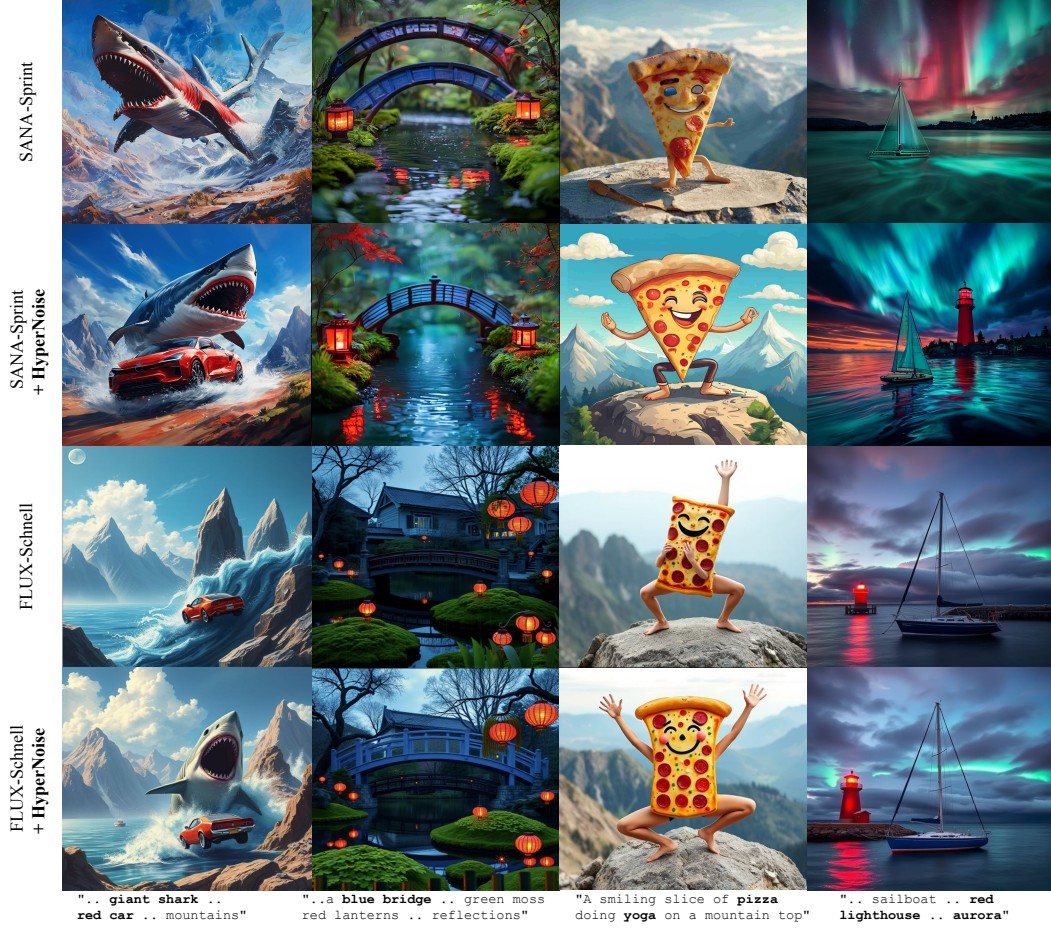

Figure 1: The same initial random noise is used for the base generation and the initialization of the noise hypernetwork. **HyperNoise** significantly improves upon the initially generated image with respect to both prompt faithfulness and aesthetic quality for both SANA-Sprint and FLUX-Schnell.

59, 93]. While both strategies can effectively boost output quality, they introduce considerable latency (exceeding 10 minutes for one generation), severely limiting their practical utility, particularly for real-time applications. This is an instantiation of a global problem of test-time scaling methods that we seek to tackle in this work. The core hypothesis of our work is whether *it is possible to capture a portion of test-time scaling benefits by integrating this knowledge into a neural network during training time?*

To address this, one might consider directly fine-tuning the diffusion model using reward signals [12, 14, 18, 53, 74, 89, 91, 108] or with Direct Preference Optimization (DPO) [34, 45, 52, 77, 98]. The objective here can be formulated as learning a *tilted distribution* (Equation 3), which upweights samples with high reward while maintaining fidelity to a base model's distribution. These methods are usually expensive to train due to the need for backpropagation through the sampling process. Instead, one might consider directly fine-tuning a step-distilled generative model to learn this target distribution. However, this approach typically involves a KL regularization to the base model that is intractable for distilled models. An imbalance or poor estimation of this can lead to the model "reward-hacking", superficially maximizing the reward metric while significantly deviating from the desired underlying data distribution, thus not achieving the genuine desired improvements.

In this work, we propose a different path to realize the benefits of the target tilted distribution (Equation 3), particularly for step-distilled generative models. Our core hypothesis is that instead of modifying the parameters of the base generator, we can achieve the desired output distribution by learning to predict an optimal initial noise distribution. We first show that such an optimal *tilted noise distribution* $p_0^\star$ exists (characterized by Equation 5). When samples from this $p_0^\star$ are passed through the frozen generator, they naturally produce outputs that are distributed according to the target data-space tilted distribution. To learn this tilted noise distribution, we introduce a lightweight

network, $f_\phi$, that transforms standard Gaussian noise into a modulated, improved noise latent. The crucial advantage of this approach lies in its optimization objective. In particular, the regularization term, a KL divergence between the modulated noise distribution and the standard Gaussian prior, is defined entirely in the noise space. We show that this noise-space KL divergence can be made tractable and effectively approximated by an $L_2$ penalty on the magnitude of the noise modification.

This lightweight network forms the core of our approach, which we term **Noise Hyper**networks. It functions akin to a hypernetwork [2, 30, 31, 39, 63, 72, 95, 105] as rather than generating the final image, it produces a specific, optimized starting latent for the main frozen generative model. This effectively guides the output of the base model without any changes to its parameters. Broadly, a hypernetwork is an auxiliary model trained to generate crucial inputs or parameters of a primary model. Our $f_\phi$ embodies this concept by learning to predict the optimized initial noise as input to the frozen generator. Consequently, *our proposed approach is effectively training a Noise Hypernetwork to perform the task of test-time noise optimization*, by learning to directly output an optimized noise latent in a single step sidestepping the need for expensive, iterative test-time optimization.

Our practical implementation of the noise hypernetwork utilizes Low-Rank Adaptation (LoRA), ensuring it remains parameter-efficient and adds negligible computational cost during inference. We apply our method to text-to-image generation, conducting evaluations with an illustrative "redness" reward task to demonstrate core mechanics, as well as complex alignments using sophisticated human-preference reward models. We demonstrate the efficacy of our approach by applying it to distilled diffusion models SD-Turbo [83], SANA-Sprint [13], and FLUX-Schnell. Overall, our experiments show that we can recover a substantial portion of the quality gains from explicit test-time optimization at a fraction of the computational inference cost. In summary, our contributions are:

1. We introduce HyperNoise, a novel framework that learns to predict an optimized initial noise for a fixed distilled generator, effectively moving test-time noise optimization benefits and computational costs into a one-time post-training stage.
2. We propose the first theoretically grounded framework for learning the reward tilted distribution of distilled generators, through a tractable noise-space objective that maintains fidelity to the base model while optimizing for desired characteristics.
3. We demonstrate through extensive experiments significant enhancements in generation quality for state-of-the-art distilled models with minimal added inference latency, making high-quality, reward-aligned generation practical for fast generators.

## 2 Background

**Preliminaries.** Recent generative models are based on a time-dependent formulation between a standard Gaussian distribution $\mathbf{x}_0 \sim p_0 = \mathcal{N}(0, \mathbf{I})$ and a data distribution $\mathbf{x}_1 \sim p_{data}$. These models define an interpolation between the initial noise and the data distribution, such that

$$\mathbf{x}_t = \alpha_t \mathbf{x}_0 + \sigma_t \mathbf{x}_1, \tag{1}$$

where $\alpha_t$ is a decreasing and $\sigma_t$ is an increasing function of $t \in [0, 1]$. Score-based diffusion [33, 43, 47, 85, 86] and flow matching [3, 55, 56] models share the observation that the process $\mathbf{x}_t$ can be sampled dynamically using a stochastic or ordinary differential equation (SDE or ODE). The neural networks parameterizing these ODEs/SDEs are trained to learn the underlying dynamics, typically by predicting the score of the perturbed data distribution or the conditional vector field. Generating a sample then involves simulating this learned differential equation starting from $\mathbf{x}_0 \sim p_0$.

**Step-Distilled Models.** The iterative simulation of such ODEs/SDEs often requires numerous steps, leading to slow sample generation. To address this latency, distillation techniques have emerged as a powerful approach. The objective is to train a "student" model that emulates the behavior of a pre-trained "teacher" model (multi-step ODE/SDE simulation) but achieves this with drastically fewer, or even a single, evaluation step(s). Prominent distillation methods such as Adversarial Diffusion Distillation [83], Consistency Models [58, 87], or Flow Maps [9, 10] have enabled the development of highly efficient few-step generative models, like SD/SDXL-Turbo [83] and SANA-Sprint [13]. In this work, we denote such a distilled generator by $g_\theta$. The significantly reduced number of sampling steps in these distilled models makes them more amenable to various optimization techniques and practical for real-time applications, which is why they are the focus of our work.

**Test-Time Noise Optimization.** Test-time optimization techniques aim to improve pre-trained generative models on a per-sample basis at inference. One prominent gradient-based strategy is

test-time noise optimization [6, 29, 46, 67, 90, 97]. Given a pre-trained generator $g_\theta$ (which could be a multi-step diffusion or flow matching model), this approach optimizes the initial noise $\mathbf{x}_0$ for each generation instance. The objective is to find an improved $\mathbf{x}_0^\star$ that maximizes a given reward $r(g_\theta(\mathbf{x}_0))$, often subject to regularization and can be formulated as

$$\mathbf{x}_0^\star = \arg\max_{\mathbf{x}_0}(r(g_\theta(\mathbf{x}_0)) - \mathrm{Reg}(\mathbf{x}_0)), \tag{2}$$

where $\mathrm{Reg}(\mathbf{x}_0)$ is a regularization term designed to keep $\mathbf{x}_0^\star$ within a high-density region of the prior noise distribution $p_0$, thus ensuring the generated sample $g_\theta(\mathbf{x}_0^\star)$ remains plausible. ReNO [22] adapted this framework for distilled generators $g_\theta$, enabling more efficient test-time optimization compared to full diffusion models. However, this per-sample optimization still incurs significant computational costs at inference, involving multiple forward and backward passes, and increased GPU memory. This inherent latency and computational burden motivate the exploration of methods that can imbue models with desired properties without per-instance test-time optimization.

**Reward-based Fine-tuning and the Tilted Distribution.** To circumvent the per-sample inference costs associated with test-time optimization, an alternative paradigm is to directly fine-tune the generative model $g_\theta$ to align with a reward function. We consider the pre-trained base distilled diffusion model $g_\theta$, which transforms an initial noise sample $\mathbf{x}_0$ into an output sample $\mathbf{x} = g_\theta(\mathbf{x}_0)$. The distribution of these generated output samples is the pushforward of $p_0$ by $g_\theta$, which we denote as $p^{\mathrm{base}} = (g_\theta)_\sharp p_0$. Given $g_\theta$ and a differentiable reward function $r(\mathbf{x}) : \mathbb{R}^d \to \mathbb{R}$ that quantifies the preference of samples $\mathbf{x}$, our objective is to learn the so called *tilted distribution*

$$p^\star(\mathbf{x}) \propto p^{\mathrm{base}}(\mathbf{x})\exp(r(\mathbf{x})). \tag{3}$$

This target distribution is defined to upweight samples with high rewards under $r(\mathbf{x})$ while staying close to the original $p^{\mathrm{base}}(\mathbf{x})$. We would like to learn $p^\star(\mathbf{x})$ by minimizing the KL divergence $D_{\mathrm{KL}}(p^\phi \| p^\star)$. Here, $p^\phi$ is the distribution generated by modifying the base process using trainable parameters $\phi$. e.g. $\phi$ could correspond to a fine-tuned version of $\theta$. This objective can be decomposed such that

$$\min_\phi D_{\mathrm{KL}}(p^\phi \| p^\star) = \min_\phi D_{\mathrm{KL}}(p^\phi \| p^{\mathrm{base}}) - \mathbb{E}_{\mathbf{x}\sim p^\phi}[r(\mathbf{x})], \tag{4}$$

where we omit the normalization constant of $p^\star(\mathbf{x})$ which is constant w.r.t. $\phi$ (see Appendix A.2). This objective encourages the learned model $p^\phi$ to generate high-reward samples while regularizing its deviation from the original base distribution $p^{\mathrm{base}}$.

**Challenges in Direct Reward Fine-tuning of Distilled Models.** Directly optimizing Equation 4 by fine-tuning the parameters of a distilled, e.g. one-step, $g_\theta$ poses significant challenges. The term $D_{\mathrm{KL}}(p^\phi \| p^{\mathrm{base}})$ requires evaluating the densities of $p^\phi$ and $p^{\mathrm{base}}$. For typical neural network generators, these densities involve Jacobian determinants through the change-of-variable formula, which are often intractable or computationally prohibitive to compute for high-dimensional data [70]. Previously, a line of work has analyzed fine-tuning Diffusion [89, 91] and Flow matching [18] models based on Equation 4 through the lens of Stochastic Optimal Control. However, this formulation relies on dynamical generative models (SDEs) and its application to distilled models is not straightforward, as these often lack the explicit continuous-time dynamical structure (e.g., an underlying SDE or ODE) that these fine-tuning techniques leverage.

## 3 Noise Hypernetworks

Given the challenges in directly fine-tuning $g_\theta$, we introduce Noise Hypernetworks (**HyperNoise**), a novel theoretically grounded approach to learn $p^\star$ for distilled generative models. The core idea is to learn a new distribution for the initial noise, $p_0^\phi$, such that samples $\hat{\mathbf{x}}_0 \sim p_0^\phi$, when passed through the fixed generator $g_\theta$, produce outputs $\mathbf{x} = g_\theta(\hat{\mathbf{x}}_0)$ that are effectively drawn from the target tilted distribution $p^\star(\mathbf{x})$ (Equation 3). Instead of modifying the parameters $\theta$ of the base generator, we keep $g_\theta$ fixed. This requires $p_0^\phi$ to approximate an optimal modulated noise distribution, $p_0^\star$. This *tilted noise distribution*, which precisely steers $g_\theta$ to $p^\star$, can be characterized by (Appendix A.3)

$$p_0^\star(\mathbf{x}_0) \propto p_0(\mathbf{x}_0)\exp(r(g_\theta(\mathbf{x}_0))). \tag{5}$$

To realize the modulated noise distribution $p_0^\phi$, we parameterize it using a learnable noise hypernetwork $f_\phi$ (with parameters $\phi$). This network defines a transformation $T_\phi$ that maps initial noise

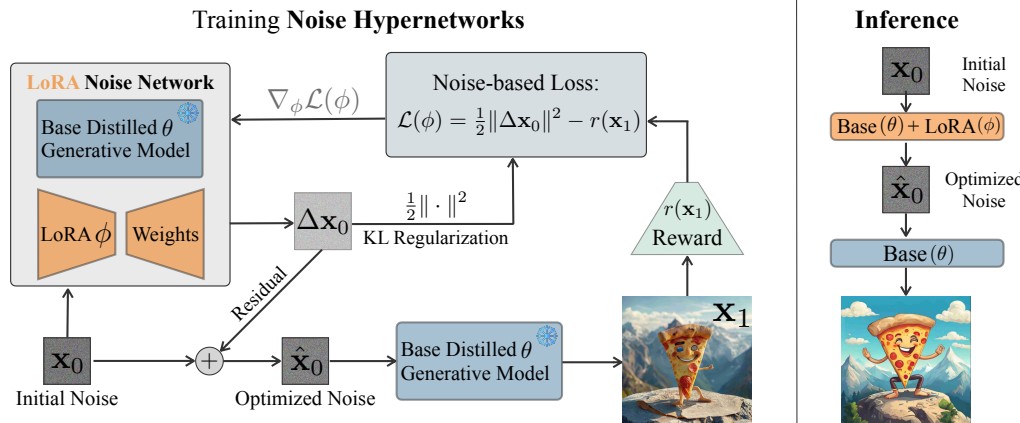

Figure 2: Illustration of our proposed HyperNoise approach. During training, the LoRA parameters are trained to predict improved noises and are optimized by reward maximization subject to KL regularization. During inference, the noise hypernetwork directly predicts the improved noise initialization which is used for the final generation.

samples $\mathbf{x}_0 \sim p_0$ to modulated samples $\hat{\mathbf{x}}_0$ via a residual formulation such that

$$\hat{\mathbf{x}}_0 = T_\phi(\mathbf{x}_0) \coloneqq \mathbf{x}_0 + f_\phi(\mathbf{x}_0). \tag{6}$$

The distribution of these modulated samples, $p_0^\phi$, is thus the pushforward of $p_0$ by $T_\phi$, i.e., $p_0^\phi = (T_\phi)_\sharp p_0$. We propose to train the parameters $\phi$ of the noise modulation network $f_\phi$ by minimizing the KL divergence $D_{\mathrm{KL}}(p_0^\phi \| p_0^\star)$. This can be shown to be equivalent to minimizing the loss function

$$\mathcal{L}_{\mathrm{noise}}(\phi) = D_{\mathrm{KL}}(p_0^\phi \| p_0) - \mathbb{E}_{\hat{\mathbf{x}}_0 \sim p_0^\phi}[r(g_\theta(\hat{\mathbf{x}}_0))]. \tag{7}$$

Analogously to Equation 4, this objective encourages $p_0^\phi$ (and thus $f_\phi$) to produce initial noise samples $\hat{\mathbf{x}}_0$ that effectively steer the fixed generator $g_\theta$ towards high-reward outputs $\mathbf{x}$. The KL term $D_{\mathrm{KL}}(p_0^\phi \| p_0)$ regularizes this steering by ensuring $p_0^\phi$ remains close to the original noise distribution $p_0$. Next, we show that in contrast to Equation 4, $\mathcal{L}_{\mathrm{noise}}$ can be made tractable.

## 3.1 KL Divergence in Noise Space

The resulting KL divergence term $D_{\mathrm{KL}}(p_0^\phi \| p_0)$ is derived in detail in Appendix A. The derivation involves the change of variables formula, simplification of Gaussian log-PDF terms, and an application of Stein's Lemma. This leads to the following expression for the KL divergence:

$$D_{\mathrm{KL}}(p_0^\phi \| p_0) = \mathbb{E}_{\mathbf{x}_0 \sim p_0}[\tfrac{1}{2}\|f_\phi(\mathbf{x}_0)\|^2 + \mathrm{Tr}(J_{f_\phi}(\mathbf{x}_0)) - \log|\det(I + J_{f_\phi}(\mathbf{x}_0))|], \tag{8}$$

where $J_{f_\phi}(\mathbf{x}_0)$ is the Jacobian of $f_\phi$ with respect to $\mathbf{x}_0$. Let $\mathcal{E}(A) \coloneqq \mathrm{Tr}(A) - \log|\det(I + A)|$. Then Equation 8 can be rewritten as $D_{\mathrm{KL}}(p_0^\phi \| p_0) = \mathbb{E}_{\mathbf{x}_0 \sim p_0}[\tfrac{1}{2}\|f_\phi(\mathbf{x}_0)\|^2 + \mathcal{E}(J_{f_\phi}(\mathbf{x}_0))]$. To simplify this expression, we analyze the error term $\mathcal{E}(J_{f_\phi}(\mathbf{x}_0))$. The following Theorem provides a bound on this term under a Lipschitz assumption on $f_\phi$.

**Theorem 1** (Bound on Log-Determinant Approximation Error). *Let $A = J_{f_\phi}(\mathbf{x}_0)$ be the $d \times d$ Jacobian matrix of $f_\phi(\mathbf{x}_0)$. Assume $f_\phi$ is L-Lipschitz continuous, such that its Lipschitz constant $L < 1$. This implies that the spectral radius $\rho(A) \leq L < 1$. Then, the error term $\mathcal{E}(A) = Tr(A) - \log|\det(I + A)|$ is bounded by*

$$|\mathcal{E}(A)| \leq d(-\log(1 - L) - L). \tag{9}$$

See Appendix A.4 for the full proof. Theorem 7 shows that if the Lipschitz constant $L$ of $f_\phi$ is sufficiently small (specifically, $L < 1$), the error term $|\mathcal{E}(A)|$ is bounded. For small $L$, $-\log(1-L) - L \approx L^2/2$, making the bound approximately $dL^2/2$. Thus, the expected error $\mathbb{E}_{\mathbf{x}_0 \sim p_0}[\mathcal{E}(J_{f_\phi}(\mathbf{x}_0))]$ becomes negligible if $L$ is kept small. Under this condition, we can approximate the KL divergence with

$$D_{\mathrm{KL}}(p_0^\phi \| p_0) \approx \mathbb{E}_{\mathbf{x}_0 \sim p_0}[\tfrac{1}{2}\|f_\phi(\mathbf{x}_0)\|^2]. \tag{10}$$

This approximation simplifies the KL divergence term in our objective to a computationally tractable $L_2$ penalty on the magnitude of the noise modification $f_\phi(\mathbf{x}_0)$. Substituting it into our initial noise modulation objective (Equation 7), we arrive at the final loss to minimize

$$\mathcal{L}_{\text{noise}}(\phi) = \mathbb{E}_{\mathbf{x}_0 \sim p_0}[\tfrac{1}{2}\|f_\phi(\mathbf{x}_0)\|^2 - r(g_\theta(\mathbf{x}_0 + f_\phi(\mathbf{x}_0)))]. \tag{11}$$

**Connection to test-time noise optimization.** Our proposed method addresses the same fundamental goal as Noise Optimization (Equation 2) of steering generation towards high-reward outputs while maintaining fidelity to the base distribution. However, instead of performing iterative optimization for each sample at inference time, we amortizes this optimization into a one-time post-training process. By learning the noise modulation network $f_\phi$, we effectively pre-computes a general policy for transforming any initial noise $\mathbf{x}_0$. Consequently, steered generation with HyperNoise remains highly efficient at inference, requiring only a single forward pass through $f_\phi$ and then $g_\theta$.

**Theoretical Justification via Data Processing Inequality.** The KL divergence term $D_{\text{KL}}(p_0^\phi \| p_0)$ in our objective (Equation 7) provides a principled way to regularize the output distribution in data space. The Data Processing Inequality (DPI) [15] states that for any function, such as our fixed generator $g_\theta$, the KL divergence between its output distributions is upper-bounded by the KL divergence between its input distributions. In our context, where $p_0^\phi = (T_\phi)_\sharp p_0$ is the distribution of modulated noise $\hat{\mathbf{x}}_0 = T_\phi(\mathbf{x}_0)$ and $p^{\text{base}} = (g_\theta)_\sharp p_0$ is the base output distribution, the DPI implies

$$D_{\text{KL}}(p_0^\phi \| p_0) \geq D_{\text{KL}}((g_\theta)_\sharp p_0^\phi \| (g_\theta)_\sharp p_0). \tag{12}$$

Thus, by minimizing $D_{\text{KL}}(p_0^\phi \| p_0)$ in the noise space, we effectively minimizes an upper bound on the KL divergence between the steered output distribution $(g_\theta)_\sharp p_0^\phi$ and the original base distribution $p^{\text{base}}$. This offers a theoretically grounded mechanism for controlling the deviation of the generated data distribution, complementing the empirical reward maximization, even when direct computation of data-space KL divergences (as in Equation 4) is intractable.

## 3.2 Effective Implementation

To implement Noise Hypernetworks efficiently and ensure stable training, we adopt several key strategies for the noise modulation network $f_\phi$ and the training process, summarized in Algorithm 1. Note that our training algorithm (Equation 11) does not require target data samples from $p^\star$, $p^{base}$, nor $p_{data}$. It only requires: (1) base noise samples $\mathbf{x}_0 \sim p_0$, (2) the fixed generator $g_\theta$, and (3) the reward function $r(\cdot)$. For conditional $g_\theta(\cdot|c)$, it additionally requires the conditions $c$.

**Lightweight Noise Hypernetwork with LoRA.** The noise modulation network $f_\phi$ is instantiated by reusing the architecture of the pre-trained generator $g_\theta$ and making it trainable via Low-Rank Adaptation (LoRA) [35]. The original $g_\theta$ weights are frozen, and only the LoRA adapter parameters in $f_\phi$ are learned. This approach is parameter-efficient, reducing memory and computational overhead as we only need to keep $g_\theta$ in memory once. It also allows $f_\phi$ to inherit useful inductive biases from $g_\theta$'s architecture. For conditional models $g_\theta(\cdot|c)$, $f_\phi(\mathbf{x}_0|c)$ can similarly leverage learned conditional representations by applying LoRA to conditioning pathways, e.g. the learned text-conditioning of a text-to-image model.

---

**Algorithm 1 HyperNoise**

---

1: **Input:** $g_\theta$ (distilled generative Model), $r$ (reward fn), Optional $\mathcal{C} = \{c_i\}_{i=1}^N$ (condition dataset)
2: Initialize Noise Hypernetwork $f_\phi(\cdot) = \mathbf{0}$ through LoRA weights $\phi$ applied on top of $g_\theta$
3: **while** training **do**
4:     Sample noise $\mathbf{x}_0 \sim \mathcal{N}(0, \mathbf{I})$, $c = \varnothing$
5:     **if** $\mathcal{C}$ **then**
6:         Sample condition $c \sim \mathcal{C}$
7:     Predict modulated noise $\Delta \mathbf{x}_0 = f_\phi(\mathbf{x}_0, c)$
8:     Generate $\mathbf{x}_1 = g_\theta(\mathbf{x}_0 + \Delta \mathbf{x}_0, c)$
9:     Compute Loss $\mathcal{L}_{\text{noise}}(\phi) = \frac{1}{2}\|\Delta \mathbf{x}_0\|^2 - r(\mathbf{x}_1)$
10:     Gradient step on $\nabla_\phi \mathcal{L}_{\text{noise}}(\phi)$
11: **return** Noise Hypernetwork LoRA weights $\phi$

---

**Initialization.** We propose to initialize $f_\phi$ such that its output $f_\phi(\cdot) = \mathbf{0}$. This is crucial for training stability and supports the validity of the $L_2$ approximation for $D_{\text{KL}}(p_0^\phi \| p_0)$ (Equation 10) from the start of training. Specifically, we modify the final layer of $f_\phi$ to output only the LoRA-generated perturbation, which are initialized to output $\mathbf{0}$ (this is achieved by setting the second LoRA matrix, often denoted $B$, to zero), *without* using any frozen base weights. This ensures that at initialization $f_\phi(\cdot) = \mathbf{0}$ such that effectively $\hat{\mathbf{x}}_0 = \mathbf{x}_0 + f_\phi(\mathbf{x}_0) = \mathbf{x}_0$, making $p_0^\phi = p_0$.

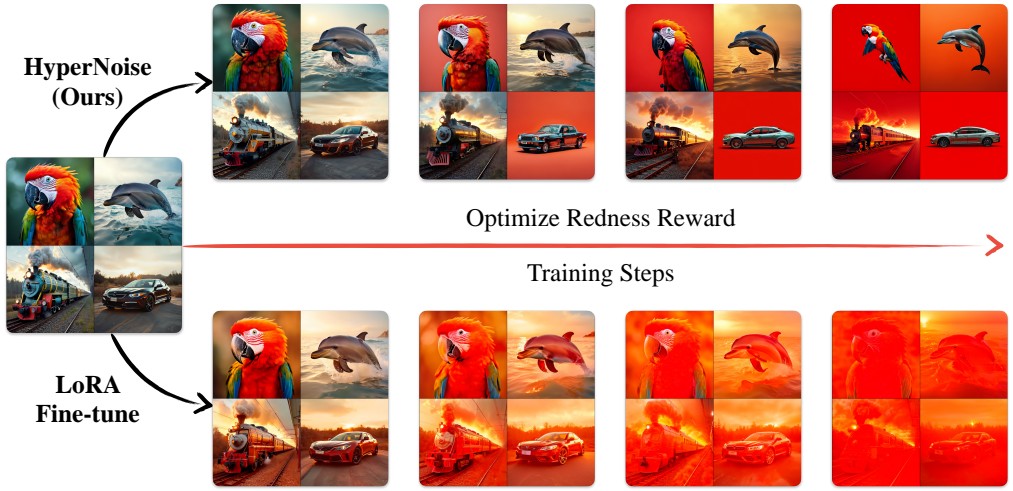

HyperNoise (Ours)

Optimize Redness Reward

Training Steps

LoRA Fine-tune

Figure 3: An illustrative example of optimizing for learning the tilted distribution with an image redness reward. We show direct LoRA fine-tuning of SANA-Sprint [13] in comparison to training a noise hypernetwork with our proposed objective. Notably, when training with our objective, the model optimizes the desired reward while staying considerably closer to $p^{\text{base}}$, as showcased by the model not diverging from the image manifold, unlike in direct LoRA fine-tuning.

## 4 Experiments

Our experimental evaluation is designed to assess the efficacy of our objective for the popular setting of text-to-image (T2I) models. We benchmark the noise hypernetwork against established methods, primarily direct LoRA fine-tuning of the base generative model [74], and investigate its capacity to match or recover the performance gains typically associated with test-time scaling techniques like ReNO [22], but through a post-training approach. To clearly delineate these comparisons, we structure our experiments as follows: We first present an illustrative experiment employing a *"redness reward"*. This controlled setting is designed to demonstrate the advantages of our training objective, particularly its ability to optimize for a target reward while mitigating divergence from the base model's learned data manifold $p^{\text{base}}$. Subsequently, we extend our evaluation to more complex and practical scenarios, focusing on aligning generative models with *human-preference reward models*.

### 4.1 Redness Reward

We begin our evaluation with the goal of learning the tilted distribution (Equation 3) given a redness reward. This metric helps showcase the potential underlying issue of directly fine-tuning the generation model $g_\phi$ (a fine-tuned variant of the base model $g_\theta$). For this experiment, the redness reward $r(\mathbf{x})$ is defined as the difference between the red channel intensity and the average of the green and blue channel intensities: $r(\mathbf{x}) = \mathbf{x}^0 - \frac{1}{2}(\mathbf{x}^1 + \mathbf{x}^2)$, where $\mathbf{x}^i$ denotes the $i$-th color channel of the generated image $\mathbf{x}$ and is used to train the recent SANA-Sprint [13] model, for full details see Appendix B.1.

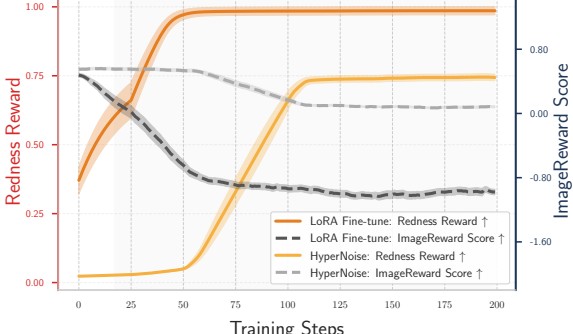

Figure 4: Trade-off between the redness reward objective and an image quality metric, ImageReward, for direct fine-tuning and Noise Hypernetworks. As opposed to direct fine-tuning, our proposed method optimizes the redness objective while not significantly dropping image quality as indicated by the ImageReward score.

The primary concern with directly fine-tuning $g_\phi$ to maximize a reward is the risk of significant deviation from the original data distribution $p^{\text{base}}$. This deviation can lead to a high

Table 1: **Quantitative Results on GenEval**. Our Noise Hypernetwork combined with (1) SD-Turbo [83], (2) SANA-Sprint 0.6B [13], and Flux-Schnell consistently improving results while maintaining few-step denoising, fast inference, and minimal memory overhead. Results from best-of-n sampling [44], ReNO [22], and prompt optimization [4, 61] are greyed out to provide a reference upper-bound in terms of applying optimization at inference. Prompt optimization † additionally requires a significant amount of calls to an LLM, either locally or through an API.

| Model | Params (B) | Time (s) ↓ | Mean ↑ | Single↑ | Two↑ | Counting↑ | Colors↑ | Position↑ | Attribution↑ |
|---|---|---|---|---|---|---|---|---|---|
| SD v2.1 [79] | 0.8 | 1.9 | 0.50 | 0.98 | 0.51 | 0.44 | 0.85 | 0.07 | 0.17 |
| SDXL [73] | 2.6 | 6.9 | 0.55 | 0.98 | 0.74 | 0.39 | 0.85 | 0.15 | 0.23 |
| DPO-SDXL [98] | 2.6 | 6.9 | 0.59 | 0.99 | 0.84 | 0.49 | 0.87 | 0.13 | 0.24 |
| Hyper-SDXL [78] | 2.6 | 0.3 | 0.56 | 1.00 | 0.76 | 0.43 | 0.87 | 0.10 | 0.21 |
| Flux-dev | 12.0 | 23.0 | 0.68 | 0.99 | 0.85 | 0.74 | 0.79 | 0.21 | 0.48 |
| SD3-Medium [21] | 2.0 | 4.4 | 0.70 | 1.00 | 0.90 | 0.72 | 0.87 | 0.31 | 0.66 |
| SD-Turbo [83] | 0.8 | 0.2 | 0.49 | 0.99 | 0.51 | 0.38 | 0.85 | 0.07 | 0.14 |
| **+ HyperNoise** | 1.1 | 0.3 | 0.57 | 0.99 | 0.65 | 0.50 | 0.89 | 0.14 | 0.22 |
| + Prompt Optimization [4, 61] | 0.8† | 95.0† | 0.59 | 0.99 | 0.76 | 0.53 | 0.88 | 0.10 | 0.28 |
| + Best-of-N [44] | 0.8 | 10.0 | 0.60 | 1.00 | 0.78 | 0.55 | 0.88 | 0.10 | 0.29 |
| + ReNO [22] | 0.8 | 20.0 | 0.63 | 1.00 | 0.84 | 0.60 | 0.90 | 0.11 | 0.36 |
| SANA-Sprint [13] | 0.6 | 0.2 | 0.70 | 1.00 | 0.80 | 0.64 | 0.86 | 0.41 | 0.51 |
| **+ HyperNoise** | 0.9 | 0.3 | 0.75 | 1.00 | 0.88 | 0.71 | 0.85 | 0.51 | 0.55 |
| + Prompt Optimization [4, 61] | 0.6† | 95.0† | 0.75 | 0.99 | 0.91 | 0.82 | 0.89 | 0.36 | 0.56 |
| + Best-of-N [44] | 0.6 | 15.0 | 0.79 | 0.99 | 0.92 | 0.72 | 0.91 | 0.53 | 0.65 |
| + ReNO [22] | 0.6 | 30.0 | 0.81 | 0.99 | 0.93 | 0.74 | 0.92 | 0.60 | 0.67 |
| FLUX-Schnell (4-step) | 12.0 | 0.7 | 0.68 | 0.99 | 0.88 | 0.66 | 0.78 | 0.27 | 0.48 |
| **+ HyperNoise** | 13.0 | 0.9 | 0.72 | 0.99 | 0.93 | 0.67 | 0.83 | 0.30 | 0.59 |
| + ReNO [22] | 12.0 | 40.0 | 0.76 | 0.99 | 0.94 | 0.70 | 0.86 | 0.39 | 0.65 |

$D_{\mathrm{KL}}(p^\phi\|p^{\mathrm{base}})$, where $p^\phi$ is the distribution induced by the fine-tuned model $g_\phi$. Such a divergence often manifests as a degradation in overall image quality or a loss of diversity, even if the target reward (e.g. redness) is achieved. Figure 4 quantitatively illustrates this trade-off by plotting the redness reward against a general image quality metric (ImageReward), comparing our Noise Hypernetwork approach with LoRA fine-tuning, while Figure 3 visually corroborates these results.

## 4.2 Human-preference Reward Models

**Implementation Details.** We conduct our primary experiments on aligning text-to-image models with human preferences using SD-Turbo [83], SANA-Sprint [13] and FLUX-Schnell. Notably, SANA-Sprint and FLUX-Schnell exhibit strong prompt-following capabilities competitive with proprietary models, making them robust base models for our evaluations. For the reward signal $r(\cdot)$ essential to our objective (Equation 11) and for the direct fine-tuning baseline, we utilize the exact same composition of reward models proposed in ReNO [22] consisting of ImageReward [103], HPSv2.1 [101], Pickscore [48], and a CLIP-score. For the noise hypernetwork, we use a LoRA [35] module on the base distilled model with the proposed initialization as described in Section 3.2. Training for the noise hypernetwork is performed using ~70k prompts from Pick-a-Picv2 [48], T2I-Compbench train set [37], and Attribute Binding (ABC-6K) [25] prompts. Our evaluations of the trained models are performed on GenEval [26], ensuring that the training and evaluation prompts do not have any overlap, measuring the generalization of the noise hypernetwork to unseen prompts. We mainly compare HyperNoise with three different test-time techniques: Best-of-N sampling [44, 59], ReNO [22], and LLM-based prompt optimization [4, 61]. As detailed in Table 1, all of these incur significantly increased computational costs at test-time, ranging from 33× to 300× slower inference compared to HyperNoise, making them impractical for large-scale deployment where efficiency is paramount. Full experimental details are provided in Appendix B.2.

**Quantitative Results.** We present our main quantitative results on the GenEval benchmark in Table 1. Our Noise Hypernetwork training scheme consistently yields significant performance gains across all model scales while maintaining near-baseline inference costs. When applied to SD-Turbo, our method nearly recovers most of the improvements from inference-time noise optimization, achieving an overall GenEval performance of 0.57 that even surpasses SDXL (which has 2× more parameters and 25× NFEs), clearly highlighting the benefits from our noise hypernetwork training. With SANA-Sprint, we observe consistent improvements (0.75 vs 0.70) over the base model, achieving the same performance as LLM-based prompt optimization while being 300× faster, and recovering about half of the performance gains achieved by ReNO with minimal GPU memory overhead. Notably, we observe

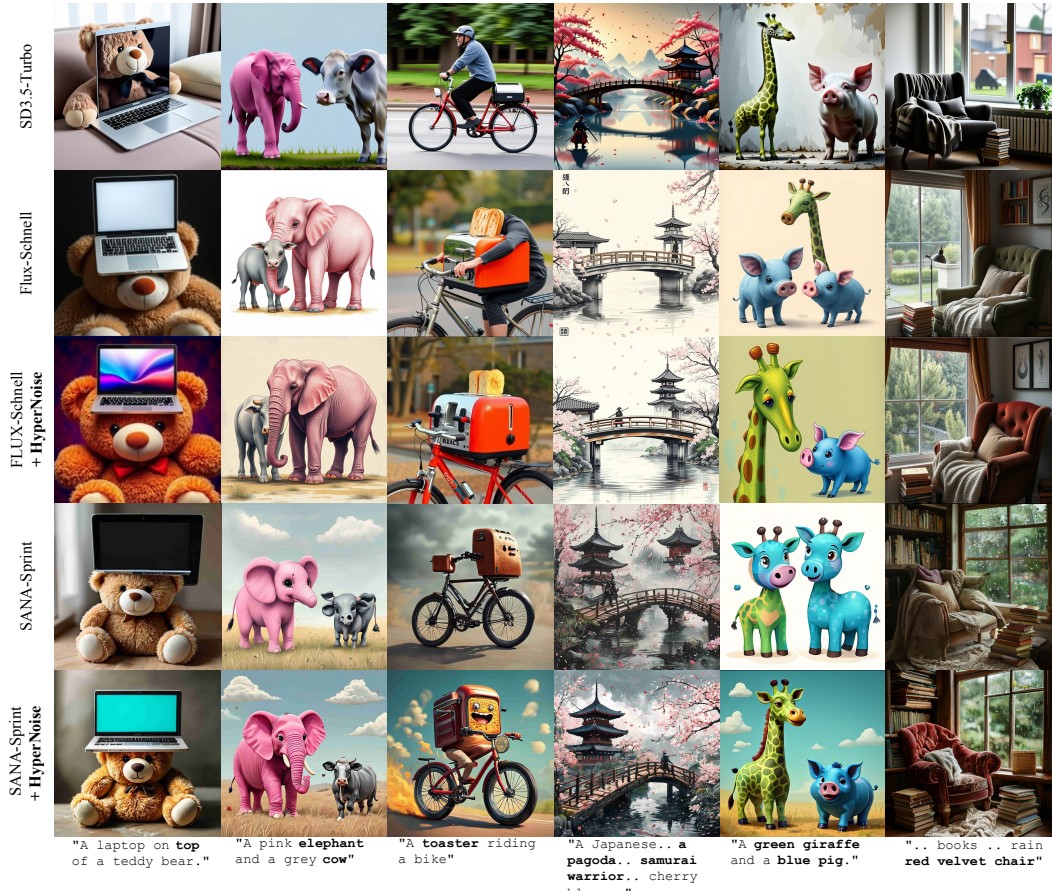

Figure 5: Qualitative comparison our proposed noise hypernetwork with popular distilled models such as Flux-Schnell, SD3.5-Turbo, SANA-Sprint for 4-step generation. Both SANA-Sprint and FLUX-Schnell share the initial noise for the base and HyperNoise generation.

similar trends for the larger 12B parameter FLUX-Schnell, where we again recover substantial performance gains (0.71 vs 0.68) while maintaining the efficiency advantages that make our approach practical for real-world deployment. The consistent efficiency gains across model scales demonstrate that our approach successfully amortizes the optimization cost during training, enabling high-quality generation without the prohibitive test-time computational overhead of alternative methods.

**Superiority over Direct Fine-tuning and Multi-Step Generalization.** In Tab. 8, we show the generalization of our training on multi-step inference despite being trained only with one-step generation. We obtain consistent improvements over SANA-Sprint for one, two, and four step generation. Notably, our model with one-step generation noticeably outperforms SANA-Sprint with 4 steps. We also illustrate how direct fine-tuning of the base model with the *same* reward objective can lead to significantly worse results, highlighting the necessity of preventing "reward-hacking" in a principled fashion. We visualize this in Appendix C.4, where we observe similar patterns as previous works for reward-hacking [14, 53, 91].

Table 2: Mean GenEval results for SANA-Sprint highlighting generalization across inference timesteps of our Noise Hypernetwork and failure of direct LoRA fine-tuning.

| SANA-Sprint [13] | NFEs | GenEval Mean↑ |
|---|---|---|
| One-step | 1 | 0.70 |
| + LoRA fine-tune [14, 74, 103] | 1 | 0.67 |
| **+ HyperNoise** | 2 | **0.75** |
| Two-step | 2 | 0.72 |
| + LoRA fine-tune [14, 74, 103] | 2 | 0.66 |
| **+ HyperNoise** | 3 | **0.76** |
| Four-step | 4 | 0.73 |
| + LoRA fine-tune [14, 74, 103] | 4 | 0.62 |
| **+ HyperNoise** | 5 | **0.77** |

**Qualitative Results.** We illustrate examples of generated images in Fig. 5 showing our method applied to both SANA-Sprint and FLUX-Schnell, alongside comparisons to SD3.5-Turbo. Our noise hypernetwork demonstrates consistent improvements across

both base models. For SANA-Sprint, the improvements are substantial: we observe both correction of generation artifacts and significantly enhanced prompt following for complex compositional requests. When applied to the already high-quality FLUX-Schnell, our method still provides noticeable improvements in detail quality and prompt adherence, demonstrating that our approach can enhance even strong base models while maintaining the efficiency advantages essential for practical deployment.

## 5 Related Work

**Test-Time Scaling.** The paradigm of test-time scaling has yielded remarkable breakthroughs, with models allocating additional computation during inference to solve increasingly complex problems. In language models, this has manifested through process reward models [60, 84, 109] and reinforcement learning from verifiable rewards [49, 64], leading to systems like o1 [40] and DeepSeek-R1 [28]. Beyond scaling denoising steps in diffusion models, test-time techniques improve generation quality by finding better initial noise or refining intermediate states during inference, often guided by pre-trained reward models. These methods fall into two categories: search-based approaches [44, 59, 92, 93] that evaluate multiple candidates, and optimization-based approaches [6, 19, 29, 46, 67, 90, 97] that iteratively refine noise or latents through gradient descent. Although both strategies achieve significant quality improvements, they introduce substantial computational overhead, with generation times frequently exceeding several minutes per image.

**Aligning Diffusion Models with Rewards.** Reward models [48, 101, 102, 103, 107] have been effectively used to directly fine-tune diffusion models using reinforcement learning [8, 12, 17, 24, 108] or direct reward fine-tuning [14, 18, 41, 50, 53, 74, 75, 103]. Alternatively, Direct Preference Optimization (DPO) [34, 45, 52, 77, 98] learns from paired comparisons rather than absolute rewards. A particular instance of reward fine-tuning [18, 89, 91] analyzes learning the reward-tilted distribution through stochastic optimal control. Uehara et al. [91] fine-tune continuous-time diffusion models by jointly optimizing both the drift term and initial noise distribution, but their SDE-based formulation requires continuous-time dynamics and backpropagation through the full sampling process, making it computationally expensive and inapplicable to step-distilled models. For distilled models, concurrent work [42, 62, 68] has explored preference tuning, though without the theoretical foundation for sampling from the target-tilted distribution that our approach provides. Wagenmaker et al. [96] apply similar noise-space optimization principles to diffusion policies in robotic control, demonstrating efficient adaptation while preserving pretrained capabilities across diverse domains.

**Hypernetworks.** Auxiliary models [30] that predict parameters of task-specific models have been used for vision [2, 31] and language tasks [39, 63, 72]. For generative models, they have been used to generate weights through diffusion [20, 99] and to speed up personalization [2, 81]. NoiseRefine [1] and Golden Noise [110] train hypernetworks to predict initial noise to replace classifier-free guidance or find reliable generations by selecting 'ground-truth' noise pairs as supervision, as opposed to the end-to-end training in our framework. Work on diffusion priors [5, 16, 23, 27, 65, 82] also adapts the noise distribution, but these approaches modify the training process rather than enabling post-hoc adaptation of pre-trained models. Concurrently, Venkatraman et al. [94] explore sampling from reward-tilted distributions for arbitrary generators, but our work demonstrates this approach at scale with comprehensive evaluation across multiple model architectures and unseen prompt distributions.

## 6 Conclusion

In this work we provide fresh perspective for post-training diffusion models through the introduction of HyperNoise, a noise modulation strategy. Our principled training objective coupled with the efficient training scheme is able to achieve a meaningful improvements in performance across multiple models while avoiding 'reward-hacking'. We hope that our efficient and effective solution for aligning diffusion models with downstream objectives finds use across a wide variety of domains and use cases, especially in cases where test-time optimization would be prohibitively expensive.

**Limitations.** Preference-tuning diffusion models heavily relies on strong pre-trained base models and meaningful reward signals. While constant improvements are made to develop better pre-trained base models, specific focus should be devoted to improving reward models that can give meaningful feedback on a variety of aspects that are important for high-quality generation.

## Acknowledgements

This work was partially funded by the ERC (853489 - DEXIM) and the Alfried Krupp von Bohlen und Halbach Foundation, which we thank for their generous support. Shyamgopal Karthik thanks the International Max Planck Research School for Intelligent Systems (IMPRS-IS) for support. Luca Eyring would like to thank the European Laboratory for Learning and Intelligent Systems (ELLIS) PhD program for support.

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

# Appendix

The Appendix is organized as follows:

## A  Theoretical Derivations

This section provides rigorous derivations for the reward-tilted noise distribution and our tractable training objective. We include a temperature parameter $\alpha > 0$ for completeness, though the main paper uses $\alpha = 1$.

### A.1  Setup and Standing Assumptions

Let $p_0(\mathbf{x}_0)$ denote the standard Gaussian density on $\mathbb{R}^d$:

$$p_0(\mathbf{x}_0) = \frac{1}{(2\pi)^{d/2}} \exp\left(-\frac{1}{2}\|\mathbf{x}_0\|^2\right). \tag{13}$$

Let $g_\theta : \mathbb{R}^d \to \mathbb{R}^d$ be the pre-trained distilled generator and $r : \mathbb{R}^d \to \mathbb{R}$ be the reward function.

**Standing Assumptions.**   Throughout this section, we assume:

1. The generator $g_\theta : \mathbb{R}^d \to \mathbb{R}^d$ is measurable.

2. The reward function $r : \mathbb{R}^d \to \mathbb{R}$ is measurable and $\mathbb{E}_{\mathbf{x}_0 \sim p_0}[e^{r(g_\theta(\mathbf{x}_0))/\alpha}] < \infty$ for our chosen temperature $\alpha > 0$.

3. For any $\mathbf{x} \in \text{Range}(g_\theta)$, the preimage set $g_\theta^{-1}(\{\mathbf{x}\})$ has a well-defined measure structure.

These assumptions are mild and realistic for neural network generators.

**Pushforward Measure and Base Distribution.**   The base generator density $p^{\text{base}}(\mathbf{x})$ is the density of the pushforward measure $(g_\theta)_\sharp P_0$, where $P_0$ is the probability measure corresponding to $p_0(\mathbf{x}_0)$. Formally, $(g_\theta)_\sharp P_0$ is defined such that for any Borel set $A \subset \mathbb{R}^d$:

$$((g_\theta)_\sharp P_0)(A) = P_0(g_\theta^{-1}(A)) = \int_{g_\theta^{-1}(A)} p_0(\mathbf{x}_0)d\mathbf{x}_0. \tag{14}$$

Under our standing assumptions, the density $p^{\text{base}}(\mathbf{x})$ can be written using the Dirac delta as:

$$p^{\text{base}}(\mathbf{x}) = \int_{\mathbb{R}^d} \delta(\mathbf{x} - g_\theta(\mathbf{x}_0))p_0(\mathbf{x}_0)d\mathbf{x}_0. \tag{15}$$

Note that in the main text, with slight abuse of notation, we write $(g_\theta)_\sharp p_0$ instead of $(g_\theta)_\sharp P_0$.

**KL Divergence.**   The Kullback-Leibler (KL) divergence between two probability densities $q(\mathbf{v})$ and $p(\mathbf{v})$ is defined as:

$$D_{\text{KL}}(q\|p) := \int_{\mathbb{R}^d} q(\mathbf{v}) \log \frac{q(\mathbf{v})}{p(\mathbf{v})}d\mathbf{v}, \tag{16}$$

provided the integral exists and is finite.

## A.2 The Reward-Tilted Output Distribution

The primary goal is to align the generator with the reward function $r(\mathbf{x})$ by targeting a *reward-tilted output distribution* $p^\star(\mathbf{x})$ that upweights high-reward samples while maintaining similarity to the base distribution.

**Definition 1** (Reward-Tilted Output Distribution). The target reward-tilted output density $p^\star(\mathbf{x})$ is defined by upweighting samples from the base generator density $p^{\text{base}}(\mathbf{x})$ according to the reward $r(\mathbf{x})$:

$$p^\star(\mathbf{x}) := \frac{1}{Z^\star} p^{\text{base}}(\mathbf{x}) \exp\left(\frac{r(\mathbf{x})}{\alpha}\right), \tag{17}$$

where $Z^\star$ is the normalization constant ensuring $p^\star(\mathbf{x})$ integrates to one:

$$Z^\star := \int_{\mathbb{R}^d} p^{\text{base}}(\mathbf{x}) \exp\left(\frac{r(\mathbf{x})}{\alpha}\right) d\mathbf{x}. \tag{18}$$

Under our standing assumptions, we have $Z^\star < \infty$. We denote $P^\star$ as the probability measure corresponding to $p^\star$.

**Interpretation.** The temperature parameter $\alpha > 0$ controls the strength of the reward signal:

- When $\alpha \to \infty$, we have $p^\star(\mathbf{x}) \to p^{\text{base}}(\mathbf{x})$ (no reward influence)
- When $\alpha \to 0$, the distribution concentrates on high-reward regions
- $\alpha = 1$ provides a natural balance between reward optimization and staying close to the base distribution

**Objective for Fine-Tuning Generator Parameters.** If we aim to fine-tune the generator parameters from $\theta$ to $\phi$, leading to a new output density $p^\phi(\mathbf{x})$ (when input is from $p_0(\mathbf{x}_0)$), a principled approach is to minimize the KL divergence $D_{\text{KL}}(p^\phi \| p^\star)$.

**Proposition 2** (KL Objective for Generator Fine-tuning). *Minimizing $D_{\text{KL}}(p^\phi \| p^\star)$ with respect to the generator parameters $\phi$ is equivalent to minimizing:*

$$J_{gen}(\phi) = D_{\text{KL}}(p^\phi \| p^{base}) - \frac{1}{\alpha} \mathbb{E}_{\mathbf{x} \sim p^\phi}[r(\mathbf{x})]. \tag{19}$$

*Proof.* Using the definition of $p^\star(\mathbf{x})$ from Equation (17):

$$
\begin{aligned}
D_{\text{KL}}(p^\phi \| p^\star) &= \int_{\mathbb{R}^d} p^\phi(\mathbf{x}) \log \frac{p^\phi(\mathbf{x})}{p^\star(\mathbf{x})} d\mathbf{x} \\
&= \int_{\mathbb{R}^d} p^\phi(\mathbf{x}) \log \frac{p^\phi(\mathbf{x}) Z^\star}{p^{\text{base}}(\mathbf{x}) \exp\left(\frac{r(\mathbf{x})}{\alpha}\right)} d\mathbf{x} \\
&= \int_{\mathbb{R}^d} p^\phi(\mathbf{x}) \left( \log \frac{p^\phi(\mathbf{x})}{p^{\text{base}}(\mathbf{x})} - \frac{r(\mathbf{x})}{\alpha} + \log Z^\star \right) d\mathbf{x} \\
&= D_{\text{KL}}(p^\phi \| p^{\text{base}}) - \frac{1}{\alpha} \mathbb{E}_{\mathbf{x} \sim p^\phi}[r(\mathbf{x})] + \log Z^\star. \tag{20}
\end{aligned}
$$

Since $\log Z^\star$ is constant with respect to $\phi$, minimizing $D_{\text{KL}}(p^\phi \| p^\star)$ is equivalent to minimizing $J_{\text{gen}}(\phi)$. $\square$

**Challenges with Direct Generator Fine-tuning.** While Proposition 2 provides a theoretically sound objective, directly optimizing it for distilled models poses significant challenges:

1. **Intractable KL term:** Computing $D_{\text{KL}}(p^\phi \| p^{\text{base}})$ requires evaluating densities of high-dimensional neural network generators, which involves intractable Jacobian determinants
2. **No continuous-time structure:** Unlike full diffusion models, distilled generators often lack explicit SDE/ODE structure that would enable techniques from stochastic optimal control

3. **Reward hacking:** Without proper regularization, optimization can lead to adversarial exploitation of the reward model, generating unrealistic samples that achieve high reward scores

These challenges motivate our alternative approach of modifying the input noise distribution while keeping the generator fixed, which we develop in the next section.

### A.3 The Reward-Tilted Noise Distribution

An alternative to modifying the generator $g_\theta$ is to modify the input noise density $p_0(\mathbf{x}_0)$ while keeping $g_\theta$ fixed. We seek an optimal *tilted noise density* $p_0^\star(\mathbf{x}_0)$ such that its pushforward through $g_\theta$ results in the target output density $p^\star(\mathbf{x})$.

**Normalization Constant in Noise Space.** First, we show that the normalization constant $Z^\star$ from Equation (18) can be expressed as an integral over the noise space. Using Equation (15) for $p^{\text{base}}(\mathbf{x})$ in the definition of $Z^\star$:

$$
\begin{aligned}
Z^\star &= \int_{\mathbb{R}^d} \exp\left(\frac{r(\mathbf{x})}{\alpha}\right) \left(\int_{\mathbb{R}^d} \delta(\mathbf{x} - g_\theta(\mathbf{x}_0'))p_0(\mathbf{x}_0')d\mathbf{x}_0'\right) d\mathbf{x} \\
&= \int_{\mathbb{R}^d} \left(\int_{\mathbb{R}^d} \exp\left(\frac{r(\mathbf{x})}{\alpha}\right) \delta(\mathbf{x} - g_\theta(\mathbf{x}_0'))d\mathbf{x}\right) p_0(\mathbf{x}_0')d\mathbf{x}_0' \quad \text{(Fubini's theorem)} \\
&= \int_{\mathbb{R}^d} \exp\left(\frac{r(g_\theta(\mathbf{x}_0'))}{\alpha}\right) p_0(\mathbf{x}_0')d\mathbf{x}_0' \quad \text{(sifting property of Dirac delta)} \\
&= \int_{\mathbb{R}^d} \exp\left(\frac{r(g_\theta(\mathbf{x}_0))}{\alpha}\right) p_0(\mathbf{x}_0)d\mathbf{x}_0.
\end{aligned}
\tag{21}
$$

**Definition 2** (Tilted Noise Distribution). The *tilted noise density* $p_0^\star(\mathbf{x}_0)$ is defined as:

$$
p_0^\star(\mathbf{x}_0) := \frac{1}{Z^\star} p_0(\mathbf{x}_0) \exp\left(\frac{r(g_\theta(\mathbf{x}_0))}{\alpha}\right),
\tag{22}
$$

where $Z^\star$ is the normalization constant from Equation (18), which by Equation (21) can be computed in noise space.

**Theorem 3** (Properties of the Tilted Noise Distribution). *Let $p_0^\star(\mathbf{x}_0)$ be the tilted noise density defined in Definition 2 and $P_0^\star$ be the corresponding probability measure. Under our standing assumptions:*

1. ***Pushforward Identity:*** *The density of the pushforward measure $(g_\theta)_\sharp P_0^\star$ is $p^\star(\mathbf{x})$.*

2. ***KL Projection:*** *The density $p_0^\star(\mathbf{x}_0)$ uniquely minimizes $D_{\text{KL}}(q_0\|p_0)$ among all noise densities $q_0(\mathbf{x}_0)$ such that the density of $(g_\theta)_\sharp Q_0$ (where $Q_0$ is the measure for $q_0$) equals $p^\star(\mathbf{x})$.*

*Proof.* **Part 1: Pushforward Identity.** We need to show that $(g_\theta)_\sharp P_0^\star$ has density $p^\star(\mathbf{x})$. For any bounded measurable set $A \subset \mathbb{R}^d$, we have:

$$
\begin{aligned}
((g_\theta)_\sharp P_0^\star)(A) = P_0^\star(g_\theta^{-1}(A)) &= \int_{g_\theta^{-1}(A)} p_0^\star(\mathbf{x}_0)d\mathbf{x}_0 \\
&= \int_{g_\theta^{-1}(A)} \frac{1}{Z^\star} p_0(\mathbf{x}_0) \exp\left(\frac{r(g_\theta(\mathbf{x}_0))}{\alpha}\right) d\mathbf{x}_0.
\end{aligned}
\tag{23}
$$

To evaluate this integral, we use the fundamental property of pushforward measures. For any measurable function $h : \mathbb{R}^d \to \mathbb{R}$:

$$
\int_{\mathbb{R}^d} h(\mathbf{x})((g_\theta)_\sharp P_0)(d\mathbf{x}) = \int_{\mathbb{R}^d} h(g_\theta(\mathbf{x}_0))P_0(d\mathbf{x}_0).
\tag{24}
$$

Applying this with $h(\mathbf{x}) = \mathbf{1}_A(\mathbf{x})\exp\left(\frac{r(\mathbf{x})}{\alpha}\right)$:

$$
\int_{g_\theta^{-1}(A)} \exp\left(\frac{r(g_\theta(\mathbf{x}_0))}{\alpha}\right) p_0(\mathbf{x}_0)d\mathbf{x}_0 = \int_A \exp\left(\frac{r(\mathbf{x})}{\alpha}\right) p^{\text{base}}(\mathbf{x})d\mathbf{x}.
\tag{25}
$$

Substituting back into Equation (23):

$$((g_\theta)_\sharp P_0^\star)(A) = \frac{1}{Z^\star} \int_A \exp\left(\frac{r(\mathbf{x})}{\alpha}\right) p^{\text{base}}(\mathbf{x})d\mathbf{x}$$

$$= \int_A \frac{1}{Z^\star} p^{\text{base}}(\mathbf{x}) \exp\left(\frac{r(\mathbf{x})}{\alpha}\right) d\mathbf{x}$$

$$= \int_A p^\star(\mathbf{x})d\mathbf{x}. \tag{26}$$

Since this holds for all measurable sets $A$, the pushforward $(g_\theta)_\sharp P_0^\star$ has density $p^\star(\mathbf{x})$.

**Part 2: KL Projection Characterization.** Consider the constrained optimization problem:

$$\min_{q_0} D_{\text{KL}}(q_0 \| p_0) \quad \text{subject to} \quad (g_\theta)_\sharp Q_0 \text{ has density } p^\star. \tag{27}$$

We use the method of Lagrange multipliers. Introduce a multiplier function $\lambda : \mathbb{R}^d \to \mathbb{R}$ and consider the functional:

$$\mathcal{L}(q_0, \lambda) = \int_{\mathbb{R}^d} q_0(\mathbf{x}_0) \log \frac{q_0(\mathbf{x}_0)}{p_0(\mathbf{x}_0)} d\mathbf{x}_0 + \int_{\mathbb{R}^d} \lambda(\mathbf{x}) \left(p^\star(\mathbf{x}) - \rho_{q_0}(\mathbf{x})\right) d\mathbf{x}, \tag{28}$$

where $\rho_{q_0}(\mathbf{x})$ is the density of $(g_\theta)_\sharp Q_0$.

For the constraint term, we can write:

$$\int_{\mathbb{R}^d} \lambda(\mathbf{x})\rho_{q_0}(\mathbf{x})d\mathbf{x} = \int_{\mathbb{R}^d} \lambda(g_\theta(\mathbf{x}_0))q_0(\mathbf{x}_0)d\mathbf{x}_0, \tag{29}$$

using the change of variables formula for pushforward measures.

Therefore:

$$\mathcal{L}(q_0, \lambda) = \int_{\mathbb{R}^d} q_0(\mathbf{x}_0) \left(\log \frac{q_0(\mathbf{x}_0)}{p_0(\mathbf{x}_0)} - \lambda(g_\theta(\mathbf{x}_0))\right) d\mathbf{x}_0 + \int_{\mathbb{R}^d} \lambda(\mathbf{x})p^\star(\mathbf{x})d\mathbf{x}. \tag{30}$$

Taking the functional derivative with respect to $q_0(\mathbf{x}_0)$ and setting to zero:

$$\frac{\delta\mathcal{L}}{\delta q_0(\mathbf{x}_0)} = \log \frac{q_0(\mathbf{x}_0)}{p_0(\mathbf{x}_0)} + 1 - \lambda(g_\theta(\mathbf{x}_0)) = 0. \tag{31}$$

This yields:

$$q_0(\mathbf{x}_0) = p_0(\mathbf{x}_0) \exp[\lambda(g_\theta(\mathbf{x}_0)) - 1]. \tag{32}$$

To satisfy the constraint, we need the density of $(g_\theta)_\sharp Q_0$ to equal $p^\star(\mathbf{x})$. Using Part 1 in reverse, this happens when:

$$q_0(\mathbf{x}_0) = \frac{1}{Z^\star} p_0(\mathbf{x}_0) \exp\left(\frac{r(g_\theta(\mathbf{x}_0))}{\alpha}\right). \tag{33}$$

Comparing with the optimality condition, we need:

$$\lambda(g_\theta(\mathbf{x}_0)) - 1 = \frac{r(g_\theta(\mathbf{x}_0))}{\alpha} - \log Z^\star. \tag{34}$$

Setting $\lambda(\mathbf{x}) = \frac{r(\mathbf{x})}{\alpha} - \log Z^\star + 1$, we obtain $q_0 = p_0^\star$.

Uniqueness follows from the strict convexity of the KL divergence in its first argument. $\qquad\square$

**Objective for Learning the Tilted Noise Distribution.** To learn a parameterized noise density $p_0^\phi(\mathbf{x}_0)$ that approximates $p_0^\star(\mathbf{x}_0)$, we minimize $D_{\text{KL}}(p_0^\phi \| p_0^\star)$.

**Proposition 4** (KL Objective for Learning Tilted Noise Density). *Minimizing $D_{\text{KL}}(p_0^\phi \| p_0^\star)$ with respect to $\phi$ is equivalent to minimizing:*

$$J_{noise}(\phi) = D_{\text{KL}}(p_0^\phi \| p_0) - \frac{1}{\alpha} \mathbb{E}_{\mathbf{x}_0 \sim p_0^\phi}[r(g_\theta(\mathbf{x}_0))]. \tag{35}$$

*Proof.* Using the definition of $p_0^\star(\mathbf{x}_0)$ from Equation (22):

$$
\begin{aligned}
D_{\mathrm{KL}}(p_0^\phi \| p_0^\star) &= \int_{\mathbb{R}^d} p_0^\phi(\mathbf{x}_0) \log \frac{p_0^\phi(\mathbf{x}_0)}{p_0^\star(\mathbf{x}_0)} d\mathbf{x}_0 \\
&= \int_{\mathbb{R}^d} p_0^\phi(\mathbf{x}_0) \log \frac{p_0^\phi(\mathbf{x}_0) Z^\star}{p_0(\mathbf{x}_0) \exp\left(\frac{r(g_\theta(\mathbf{x}_0))}{\alpha}\right)} d\mathbf{x}_0 \\
&= \int_{\mathbb{R}^d} p_0^\phi(\mathbf{x}_0) \left( \log \frac{p_0^\phi(\mathbf{x}_0)}{p_0(\mathbf{x}_0)} - \frac{r(g_\theta(\mathbf{x}_0))}{\alpha} + \log Z^\star \right) d\mathbf{x}_0 \\
&= D_{\mathrm{KL}}(p_0^\phi \| p_0) - \frac{1}{\alpha} \mathbb{E}_{\mathbf{x}_0 \sim p_0^\phi}[r(g_\theta(\mathbf{x}_0))] + \log Z^\star. \quad\quad (36)
\end{aligned}
$$

Since $\log Z^\star$ is constant with respect to $\phi$, minimizing $D_{\mathrm{KL}}(p_0^\phi \| p_0^\star)$ is equivalent to minimizing $J_{\mathrm{noise}}(\phi)$. $\qquad\square$

### A.3.1  Connection to Stochastic Optimal Control

We now show how our result connects to the sophisticated stochastic optimal control framework of Uehara et al. [91] for fine-tuning continuous-time diffusion models, demonstrating that their approach naturally reduces to our simpler result for one-step generators.

**Continuous-Time Framework.**  Uehara et al. [91] consider the entropy-regularized control problem:

$$
\max_{u,\nu} \mathbb{E}_{P^{u,\nu}}[r(\mathbf{x}_T)] - \alpha \mathbb{E}_{P^{u,\nu}} \left[ \int_0^T \frac{\|u(t, \mathbf{x}_t)\|^2}{2\sigma^2(t)} dt + \log \frac{\nu(\mathbf{x}_0)}{p_0(\mathbf{x}_0)} \right] \quad\quad (37)
$$

where $P^{u,\nu}$ is the path measure induced by the SDE with drift $f(t,\mathbf{x}) + u(t,\mathbf{x})$ and $\nu$ the initial distribution to optimize.

**Reduction to One-Step Generators.**  For a one-step generator $\mathbf{x} = g_\theta(\mathbf{x}_0)$, the stochastic process degenerates:

- The evolution is deterministic: $\mathbf{x}_T = g_\theta(\mathbf{x}_0)$
- No drift control is needed: optimal $u \equiv 0$
- Only the initial distribution $\nu$ requires optimization

The objective reduces to:

$$
\max_\nu \mathbb{E}_{\mathbf{x}_0 \sim \nu}[r(g_\theta(\mathbf{x}_0))] - \alpha \cdot D_{\mathrm{KL}}(\nu \| p_0) \quad\quad (38)
$$

**Optimal Initial Distribution.**  According to their Corollary 2, the optimal initial distribution is:

$$
\nu^*(\mathbf{x}_0) = \frac{\exp(v_0^*(\mathbf{x}_0)/\alpha) \cdot p_0(\mathbf{x}_0)}{Z^\star} \qu\quad (39)
$$

where $v_0^*(\mathbf{x}_0)$ is the value function at time $t = 0$.

**Value Function for Deterministic Generators.**  From their Lemma 1 (Feynman-Kac formulation), the value function satisfies:

$$
\exp\left(v_0^*(\mathbf{x}_0)/\alpha\right) = \mathbb{E}_{P^{0,\nu}}\left[ \exp\left(\frac{r(\mathbf{x}_T)}{\alpha}\right) \Big| \mathbf{x}_0 \right] \qu\quad (40)
$$

For the deterministic generator $g_\theta$:

$$
\begin{aligned}
\mathbb{E}[\exp(r(\mathbf{x}_T)/\alpha)|\mathbf{x}_0] &= \mathbb{E}[\exp(r(g_\theta(\mathbf{x}_0))/\alpha)|\mathbf{x}_0] \\
&= \exp(r(g_\theta(\mathbf{x}_0))/\alpha) \quad \text{(deterministic given } \mathbf{x}_0) \quad\quad (41)
\end{aligned}
$$

Therefore: $v_0^*(\mathbf{x}_0) = r(g_\theta(\mathbf{x}_0))$.

**Final Result and Validation.** Substituting back into the optimal distribution formula:

$$\nu^*(\mathbf{x}_0) = \frac{\exp(r(g_\theta(\mathbf{x}_0))/\alpha) \cdot p_0(\mathbf{x}_0)}{Z^*} \tag{42}$$

where $Z^* = \int \exp(r(g_\theta(\mathbf{x}_0))/\alpha) \cdot p_0(\mathbf{x}_0)d\mathbf{x}_0$.

This is precisely our $p_0^\star$ in Definition 2. This alignment between the two frameworks is significant, as it confirms that:

1. Our direct variational approach and the general stochastic control theory yield the same optimal noise distribution.

2. This equivalence arises because for one-step generators, the continuous-time framework naturally collapses to our setting, with their value function $v_0^\star$ simplifying to the composed reward $r \circ g_\theta$.

This connection not only validates our result but also situates it as an important special case within the broader theory of stochastic control.

## A.4 Tractable KL Divergence for Noise Modification

We derive a tractable expression for $D_{\mathrm{KL}}(p_0^\phi \| p_0)$ where $p_0^\phi$ is the density of modified noise $\hat{\mathbf{x}}_0 = T_\phi(\mathbf{x}_0)$ with $T_\phi(\mathbf{x}_0) = \mathbf{x}_0 + f_\phi(\mathbf{x}_0)$. This derivation involves the change of variables formula, simplification of Gaussian log-PDF terms, and an application of Stein's Lemma.

**Setup and Minimal Assumptions.** Let $T_\phi : \mathbb{R}^d \to \mathbb{R}^d$ be the residual transformation:

$$T_\phi(\mathbf{x}_0) = \mathbf{x}_0 + f_\phi(\mathbf{x}_0) \tag{43}$$

where $f_\phi : \mathbb{R}^d \to \mathbb{R}^d$ is a learned perturbation function with Jacobian $J_{f_\phi}(\mathbf{x}_0) = \frac{\partial f_\phi(\mathbf{x}_0)}{\partial \mathbf{x}_0^T}$.

**Assumption 1** (Regularity Conditions). We assume:

1. $f_\phi$ is continuously differentiable

2. $T_\phi$ is a global diffeomorphism (invertible with continuous derivatives)

3. $f_\phi$ satisfies the regularity conditions for Stein's lemma: $\mathbb{E}[\|f_\phi(\mathbf{x}_0)\|^2] < \infty$ and $\mathbb{E}[\|\mathbf{x}_0\|\|f_\phi(\mathbf{x}_0)\|] < \infty$ for $\mathbf{x}_0 \sim \mathcal{N}(\mathbf{0}, I)$

**Sufficient Condition for Global Diffeomorphism.** While Assumption 1 requires $T_\phi$ to be a global diffeomorphism, we provide a practical sufficient condition:

**Lemma 5** (Lipschitz Condition for Invertibility). *If $f_\phi$ is $L$-Lipschitz continuous with $L < 1$, then $T_\phi$ is a global diffeomorphism.*

*Proof.* Bi-Lipschitz bounds: for any $\mathbf{x}_0, \mathbf{x}_0'$,

$$\|T_\phi(\mathbf{x}_0) - T_\phi(\mathbf{x}_0')\| \leq \|\mathbf{x}_0 - \mathbf{x}_0'\| + \|f_\phi(\mathbf{x}_0) - f_\phi(\mathbf{x}_0')\| \leq (1 + L) \|\mathbf{x}_0 - \mathbf{x}_0'\|, \tag{44}$$

$$\|T_\phi(\mathbf{x}_0) - T_\phi(\mathbf{x}_0')\| \geq \|\mathbf{x}_0 - \mathbf{x}_0'\| - \|f_\phi(\mathbf{x}_0) - f_\phi(\mathbf{x}_0')\| \geq (1 - L) \|\mathbf{x}_0 - \mathbf{x}_0'\|. \tag{45}$$

Hence $T_\phi$ is injective. For surjectivity, fix any target $\mathbf{y}$ and define $G_\mathbf{y}(\mathbf{z}) = \mathbf{y} - f_\phi(\mathbf{z})$, a contraction with constant $L < 1$. By Banach's fixed-point theorem there exists a unique $\mathbf{z}^\star$ with $\mathbf{z}^\star = G_\mathbf{y}(\mathbf{z}^\star)$, i.e., $T_\phi(\mathbf{z}^\star) = \mathbf{y}$. Finally, $J_{T_\phi}(\mathbf{x}_0) = I + J_{f_\phi}(\mathbf{x}_0)$ is invertible for all $\mathbf{x}_0$ (its smallest singular value is at least $1 - L > 0$), and the inverse is $C^1$ by the inverse function theorem. Thus $T_\phi$ is a global $C^1$ diffeomorphism. $\square$

**KL Divergence via Change of Variables.** Under Assumption 1, we can apply the change of variables formula. The KL divergence is:

$$D_{\mathrm{KL}}(p_0^\phi \| p_0) = \mathbb{E}_{\hat{\mathbf{x}}_0 \sim p_0^\phi} \left[ \log \frac{p_0^\phi(\hat{\mathbf{x}}_0)}{p_0(\hat{\mathbf{x}}_0)} \right] \tag{46}$$

$$= \mathbb{E}_{\mathbf{x}_0 \sim p_0} \left[ \log \frac{p_0^\phi(T_\phi(\mathbf{x}_0))}{p_0(T_\phi(\mathbf{x}_0))} \right] \tag{47}$$

By the change of variables formula:

$$p_0^\phi(T_\phi(\mathbf{x}_0)) = p_0(\mathbf{x}_0)|\det(J_{T_\phi}(\mathbf{x}_0))|^{-1} \tag{48}$$

Since $J_{T_\phi}(\mathbf{x}_0) = I + J_{f_\phi}(\mathbf{x}_0)$, substituting into Equation (47):

$$D_{\mathrm{KL}}(p_0^\phi \| p_0) = \mathbb{E}_{\mathbf{x}_0 \sim p_0}\left[\log p_0(\mathbf{x}_0) - \log p_0(T_\phi(\mathbf{x}_0)) - \log|\det(I + J_{f_\phi}(\mathbf{x}_0))|\right] \tag{49}$$

**Specialization to Gaussian Base Distribution.** For $p_0(\mathbf{x}_0) = \mathcal{N}(\mathbf{0}, I)$, the log-density difference simplifies:

$$\log p_0(\mathbf{x}_0) - \log p_0(T_\phi(\mathbf{x}_0)) = -\frac{1}{2}\|\mathbf{x}_0\|^2 + \frac{1}{2}\|T_\phi(\mathbf{x}_0)\|^2 \tag{50}$$

$$= -\frac{1}{2}\|\mathbf{x}_0\|^2 + \frac{1}{2}\|\mathbf{x}_0 + f_\phi(\mathbf{x}_0)\|^2 \tag{51}$$

$$= \mathbf{x}_0^T f_\phi(\mathbf{x}_0) + \frac{1}{2}\|f_\phi(\mathbf{x}_0)\|^2 \tag{52}$$

Substituting Equation (52) into Equation (49):

$$D_{\mathrm{KL}}(p_0^\phi \| p_0) = \mathbb{E}_{\mathbf{x}_0 \sim \mathcal{N}(\mathbf{0}, I)}\left[\mathbf{x}_0^T f_\phi(\mathbf{x}_0) + \frac{1}{2}\|f_\phi(\mathbf{x}_0)\|^2 - \log|\det(I + J_{f_\phi}(\mathbf{x}_0))|\right] \tag{53}$$

**Application of Stein's Lemma.** Under the regularity conditions in Assumption 1, Stein's lemma applies:

**Lemma 6** (Stein's Lemma for Vector Fields). *Let* $\mathbf{x} \sim \mathcal{N}(\mathbf{0}, I)$ *and* $h : \mathbb{R}^d \to \mathbb{R}^d$ *satisfy* $\mathbb{E}[\|h(\mathbf{x})\|^2] < \infty$ *and* $\mathbb{E}[\|\mathbf{x}\|\|h(\mathbf{x})\|] < \infty$. *Then:*

$$\mathbb{E}[\mathbf{x}^T h(\mathbf{x})] = \mathbb{E}[Tr(J_h(\mathbf{x}))] \tag{54}$$

Applying Lemma 6 to Equation (53), we obtain:

$$D_{\mathrm{KL}}(p_0^\phi \| p_0) = \mathbb{E}_{\mathbf{x}_0 \sim p_0}\left[\frac{1}{2}\|f_\phi(\mathbf{x}_0)\|^2 + \mathrm{Tr}(J_{f_\phi}(\mathbf{x}_0)) - \log|\det(I + J_{f_\phi}(\mathbf{x}_0))|\right] \tag{55}$$

This is exactly the expression referenced in the main text.

**Log-Determinant Approximation Analysis.** Let $\mathcal{E}(A) := \mathrm{Tr}(A) - \log|\det(I + A)|$. Then Equation (55) can be rewritten as:

$$D_{\mathrm{KL}}(p_0^\phi \| p_0) = \mathbb{E}_{\mathbf{x}_0 \sim p_0}\left[\frac{1}{2}\|f_\phi(\mathbf{x}_0)\|^2 + \mathcal{E}(J_{f_\phi}(\mathbf{x}_0))\right] \tag{56}$$

To simplify this expression, we analyze the error term $\mathcal{E}(J_{f_\phi}(\mathbf{x}_0))$. The following theorem provides a bound on this term under a Lipschitz assumption on $f_\phi$.

**Theorem 7** (Bound on Log-Determinant Approximation Error). *Let* $A = J_{f_\phi}(\mathbf{x}_0)$ *be the* $d \times d$ *Jacobian matrix of* $f_\phi(\mathbf{x}_0)$. *Assume* $f_\phi$ *is* $L$-*Lipschitz continuous, such that its Lipschitz constant* $L < 1$. *This implies that the spectral radius* $\rho(A) \leq L < 1$. *Then, the error term* $\mathcal{E}(A) = Tr(A) - \log|\det(I + A)|$ *is bounded by:*

$$|\mathcal{E}(A)| \leq d(-\log(1 - L) - L) \tag{57}$$

*Proof.* Since $f_\phi$ is $L$-Lipschitz, the spectral norm of its Jacobian satisfies $\|A\|_2 \leq L$. This implies the spectral radius $\rho(A) \leq \|A\|_2 \leq L < 1$, ensuring all eigenvalues $\lambda_i(A)$ satisfy $|\lambda_i(A)| < 1$.

Since $1 + \lambda_i(A) > 0$ for all $i$, we have $\det(I + A) > 0$, so $\log|\det(I + A)| = \log\det(I + A)$.

For $\rho(A) < 1$, the matrix logarithm series converges:

$$\log\det(I + A) = \sum_{k=1}^{\infty}\frac{(-1)^{k-1}}{k}\mathrm{Tr}(A^k) \tag{58}$$

Therefore:

$$\mathcal{E}(A) = \mathrm{Tr}(A) - \sum_{k=1}^{\infty} \frac{(-1)^{k-1}}{k} \mathrm{Tr}(A^k) \tag{59}$$

$$= \sum_{k=2}^{\infty} \frac{(-1)^k}{k} \mathrm{Tr}(A^k) \tag{60}$$

Taking absolute values and using $|\mathrm{Tr}(A^k)| \leq d \cdot \rho(A)^k \leq d \cdot L^k$:

$$|\mathcal{E}(A)| \leq \sum_{k=2}^{\infty} \frac{d \cdot L^k}{k} \tag{61}$$

$$= d \left( \sum_{k=1}^{\infty} \frac{L^k}{k} - L \right) \tag{62}$$

$$= d(-\log(1 - L) - L) \tag{63}$$

$\square$

**Practical Approximation and Final Objective.**   Theorem 7 shows that if the Lipschitz constant $L$ of $f_\phi$ is sufficiently small (specifically, $L < 1$), the error term $|\mathcal{E}(A)|$ is bounded. For small $L$, $-\log(1 - L) - L \approx L^2/2$, making the bound approximately $dL^2/2$. Thus, the expected error $\mathbb{E}_{\mathbf{x}_0 \sim p_0}[\mathcal{E}(J_{f_\phi}(\mathbf{x}_0))]$ becomes negligible if $L$ is kept small. Under this condition, we can approximate the KL divergence with:

$$D_{\mathrm{KL}}(p_0^\phi \| p_0) \approx \mathbb{E}_{\mathbf{x}_0 \sim p_0} \left[ \frac{1}{2} \| f_\phi(\mathbf{x}_0) \|^2 \right] \tag{64}$$

This approximation simplifies the KL divergence term in our objective to a computationally tractable $L_2$ penalty on the magnitude of the noise modification $f_\phi(\mathbf{x}_0)$.

**Integration with Main Objective.**   Combining our approximation with Proposition 4, and substituting Equation (64) into our initial noise modulation objective, we arrive at the final loss to minimize:

$$\mathcal{L}_{\mathrm{noise}}(\phi) = \mathbb{E}_{\mathbf{x}_0 \sim p_0} \left[ \frac{1}{2} \| f_\phi(\mathbf{x}_0) \|^2 - \frac{1}{\alpha} r\big( g_\theta(\mathbf{x}_0 + f_\phi(\mathbf{x}_0)) \big) \right] \tag{65}$$

This objective balances reward maximization against the KL regularization term, providing a principled and computationally tractable approach to learning the reward-tilted noise distribution.

**Practical Implementation Considerations.**   The validity of our approximation depends on maintaining small Lipschitz constants. In practice, this is supported by:

1. **Initialization**: Setting $f_\phi(\cdot) \equiv \mathbf{0}$ ensures $\mathcal{E}(A) = 0$ initially

2. **Regularization**: The term $\frac{1}{2} \| f_\phi(\mathbf{x}_0) \|^2$ naturally penalizes large perturbations, helping maintain small eigenvalues of $J_{f_\phi}$

While we do not explicitly enforce $L < 1$ during training, these practical measures help maintain $f_\phi$ in a regime where our approximation remains accurate throughout the optimization process.

## B   Experimental and Implementation Details

In this Section we report the details for all of our experimental results. We mainly use the SANA-Sprint 0.6B [13] model, and train it using one-step generation. Additionally, we use the default guidance scale of $4.5$ for all experiments. After training, we evaluate our models using different amounts of NFEs with one forward pass of Noise Hypernetwork beforehand.

**LoRA parameterization**  We parameterize our noise hypernetwork $f_\phi$ with LoRA weights on top of the base distilled generative model. We found this to be important mainly to reuse the conditional pathways learned by the base model. This is especially important for complex conditioning, like text. Without this paramertization, which we also explored initially, we found it difficult for the noise hypernetwork to learn an effective conditioning with limit data. While larger-scale training could be a solution to this, we found this LoRA parameterization to be an efficient solution. For a condition independent reward, e.g. the redness one, it is less important to choose such a parametrization.

**Initialization**  As described in Section 3.2, we initialize the noise network to output $f_\phi(\cdot) = \mathbf{0}$ at the start of training. We implement this by setting the output of the last base layer to $\mathbf{0}$ and initializing the LoRA weights of the second LoRA weight matrix (also reffered to as B) to $0$. This effectively initializes $f_\phi(\cdot) = \mathbf{0}$. For a stable training, this initialization is important as the model $g_\theta(f_\phi(\mathbf{x}_0) + \mathbf{x}_0))$ generates meaningful images at the start of training. In that way $f_\phi$ only needs to learn how to refine $\mathbf{x}_0$.

**Memory efficient implementation.**  Section 3.2, we train our noise hypernetwork $f_\phi$ as a special LoRA version of our base model $g_\theta$, which ignores the last layer of the base model. As visualized in Figure 2, we only need to keep the base model in memory once. Thus, the GPU memory overhead is just the added LoRA weights $\phi$. Additionally, we employ Pytorch Memsave [7] to all models, which further reduces the needed GPU memory during training enabling us to use larger batch sizes. We run all experiments in `bfloat16`. Additionally, we can leverage gradient checkpointing on the first call of the model with activated LoRA parameters to further reduce memory. We use this for our FLUX-Schnell training.

## B.1   Redness Reward

For the Redness Reward, we use SANA-Sprint 0.6B [13] as the base model. We train the model with the redness reward

$$r(\mathbf{x}) = \tfrac{1}{100} * (\mathbf{x}^0 - \frac{1}{2}(\mathbf{x}^1 + \mathbf{x}^2)),$$

where $\mathbf{x}^i$ denotes the i-th color channel of $\mathbf{x}$. We use the same amount of LoRA parameters for fine-tuning and noise hypernetwork training. In general, we keep the hyperparameters for our comparison between fine-tuning and noise hypernetwork training exactly the same. Due to the sake of illustration, we lower the learning rate for fine-tuning in this case as otherwise the model collapses to generating pure red images after a few training steps. We train on 30 prompts from the GenEval [26] promptset and evaluate on the four unseen prompts ["A photo of a parrot", "A photo of a dolphin", "A photo of a train", "A photo of a car"]. After each epoch on the 30 prompts, we compute the redness reward as well as an "imageness score" for each of the 4 evaluation prompts and average. For the imageness score, we use the ImageReward [103] human-preference reward model as it was shown to correctly quantify prompt-following capabilities. We provide the full hyperparameters in Table 3. This experiment was conducted on 1 H100 GPU.

Table 3: Hyperparameters for the Redness Reward setting

|  | Fine-tuning | Noise Hypernetwork |
|---|---|---|
| Model | SANA-Sprint [13] | SANA-Sprint [13] |
| Learning rate | $1e-4$ | $1e-3$ |
| GradNorm Clipping | 1.0 | 1.0 |
| LoRA rank | 128 | 128 |
| LoRA alpha | 256 | 256 |
| Optimizer | SGD | SGD |
| Batch size | 3 | 3 |
| Training epochs | 200 | 200 |
| Number of training prompts | 30 | 30 |
| Image size | $1024 \times 1024$ | $1024 \times 1024$ |

## B.2 Human Preference Reward Models

For our large-scale experiments, we consider SD-Turbo [83] and SANA-Sprint [13] as our two base models. For SD-Turbo we generate images in $512 \times 512$ while for SANA-Sprint we generate them of size $1024 \times 1024$. The training for the noise hypernetwork is done using ~70k prompts from Pick-a-Picv2 [48], T2I-Compbench train set [37], and Attribute Binding (ABC-6K) [25] prompts. As the reward we follow ReNO [22] and use a combination of human-preference trained reward models consisting of ImageReward [103], HPSv2.1 [101], PickScore [48], and CLIP-Score [38]. To balance these, we weigh each reward model with the same weightings as proposed in ReNO [22] and employ them with the following implementation details. All training runs were conducted on 6 H100 GPUs.

**Human Preference Score v2.1 (HPSv2.1)**  HPSv2.1 [101] is an improved version of the HPS [102] model, which uses an OpenCLIP ViT-H/14 model and is trained on prompts collected from DiffusionDB [100] and other sources.

**PickScore**  PickScore also uses the same ViT-H/14 model, however is trained on the Pick-a-Pic dataset which consists of 500k+ preferences that are collected through crowd-sourced prompts and comparisons.

**ImageReward**  ImageReward [103] trains a MLP over the features extracted from a BLIP model [51]. This is trained on a dataset of images collected from the DiffusionDB [100] prompts.

**CLIPScore**  Lastly, we use CLIPScore [32, 76], which was not designed specifically as a human preference reward model. However, it measures the text-image alignment with a score between 0 and 1. Thus, it offers a way of evaluating the prompt faithfulness of the generated image that can be optimized. We use the model provided by OpenCLIP [38] with a ViT-H/14 backbone.

Table 4: Hyperparameters for the Human-preference Reward setting

|  | Noise Hypernetwork | Fine-tuning | Noise Hypernetwork | Noise Hypernetwork |
|---|---|---|---|---|
| Model | SD-Turbo [83] | SANA-Sprint [13] | SANA-Sprint [13] | FLUX-Schnell |
| Learning rate | $1e-3$ | $1e-3$ | $1e-3$ | $1e-3$ |
| GradNorm Clipping | 10.0 | 1.0 | 1.0 | 10.0 |
| LoRA rank | 128 | 128 | 128 | 128 |
| LoRA alpha | 128 | 256 | 256 | $5*128$ |
| Optimizer | SGD | SGD | SGD | SGD |
| Batch size | 18 | 18 | 18 | 7 |
| Accumulation Steps | 1 | 3 | 3 | 4 |
| Training Epochs | $\approx 25$ | $\approx 25$ | $\approx 25$ | $\approx 25$ |
| Number of training prompts | $\approx 70k$ | $\approx 70k$ | $\approx 70k$ | $\approx 70k$ |
| Image size | $512 \times 512$ | $1024 \times 1024$ | $1024 \times 1024$ | $512 \times 512$ |

**GenEval**  Our main evaluation metric is GenEval, an object-focused framework introduced by Ghosh et al. [26] for evaluating the alignment between text prompts and generated images from Text-to-Image (T2I) models. GenEval leverages existing object detection methods to perform a fine-grained, instance-level analysis of compositional capabilities. The framework assesses various aspects of image generation, including object co-occurrence, position, count, and color. By linking the object detection pipeline with other discriminative vision models, GenEval can further verify properties like object color. All the metrics on the GenEval benchmarks are evaluated using a MaskFormer object detection model with a Swin Transformer [57] backbone. Lastly, GenEval is evaluated over four seeds and reports the mean for each metric, which we follow. Note that our FLUX-Schnell differ from the ones in Eyring et al. [22] as we use `bfloat16` instead of `float16`.

## B.3 Test-time techniques

For ReNO [22], we use the default parameters as described in their paper with 50 forward passes for one image generation. For Best-of-N [44] we use $N = 50$ with the same reward ensemble for a fair comparison. For LLM-based prompt optimization [4, 61], we use the default setup from the MILS [4] repository (https://github.com/facebookresearch/MILS/blob/main/main_

`image_generation_enhancement.py`) with local Llama 3.1 8B Instruct as the LLM. The time reflected in Table 1 reflects these local LLM calls. Note that we left the GPU memory to just the base image generation model. We modify the hyperparameters to 5 prompt proposals for each LLM call and 10 iterations, such that we also end up with 50 image evaluations for a fair comparison.

# C  Additional results

In this section we report additional quantiative ablation results and further qualitative results.

## C.1  Additional Benchmarks

Here, we report further results on two more benchmarks commonly employed in the evaluation of T2I generation. Note that again, none of the prompts in the used benchmarks are part of the training data, showcasing the generalizability of the Noise Hypernetwork to unseen prompts and also that our optimization objective through human-preference reward mdoels is disentangled from these benchmarks.

**T2I-CompBench.** T2I-CompBench is a comprehensive benchmark proposed by Park et al. [71] for evaluating the compositional capabilities of text-to-image generation models. We evaluate on the Attribute binding tasks, which includes color, shape, and texture sub-categories, where the model should bind the attributes with the correct objects to generate

Table 5: **Quantitative Results on T2I-CompBench**. The Noise Hypernetwork consistently improves performance.

| SANA-Sprint 0.6B [13] | NFEs | Color ↑ | Shape↑ | Texture↑ |
|---|---|---|---|---|
| One-step | 1 | 0.72 | 0.49 | 0.63 |
| **+ Noise Hypernetwork** | 2 | **0.75** | **0.53** | **0.64** |
| Two-step | 2 | 0.73 | 0.50 | 0.64 |
| **+ Noise Hypernetwork** | 3 | **0.76** | **0.53** | 0.64 |
| Four-step | 4 | 0.73 | 0.50 | 0.64 |
| **+ Noise Hypernetwork** | 5 | **0.76** | **0.54** | **0.65** |

the complex scene. The attribute binding subtasks are evaluated using BLIP-VQA (i.e., generating questions based on the prompt and applying VQA on the generated image). We perform these evaluations on the validation set of prompts and results are shown in Tab. 5 and observe consistent improvements across steps and categories.

**DPG-Bench.** We provide results on DPG-Bench [36] in Tab. 6. Broadly, while performance increases for all models with increasing timesteps, we note that the results for the four step SANA-Sprint model is nearly matched by the one-step model with our noise hypernetwork. We also note that the DPG-Bench score of 80.82 surpasses powerful models such as SDXL [73], Pixart-Σ, and is only surpassed by much larger models such as SD3 [21], and Flux. Finally, we also note that the human-preference reward models that we utilize all have a CLIP/BLIP encoder that

Table 6: DPG-Bench results for SANA-Sprint highlighting generalization across inference timesteps of our Noise Hypernetwork.

| SANA-Sprint 0.6B [13] | NFEs | DPG-Bench Score↑ |
|---|---|---|
| One-step | 1 | 77.59 |
| **+ Noise Hypernetwork** | 2 | **79.20** |
| Two-step | 2 | 79.07 |
| **+ Noise Hypernetwork** | 3 | **79.74** |
| Four-step | 4 | 79.54 |
| **+ Noise Hypernetwork** | 5 | **80.82** |

limits the length of the captions to $< 77$ tokens, which offers minimal scope of improvements for benchmarks involving much longer prompts that exceed this context window. Future reward models that either utilize different CLIP models (e.g. Long-CLIP [104]) or LLM-based decoders (e.g. VQAScore [54]) would enable improving prompt following of these models more dramatically in the case of long prompts.

| | SANA-Sprint | + HyperNoise | + LoRA Fine-tune |
|---|---|---|---|

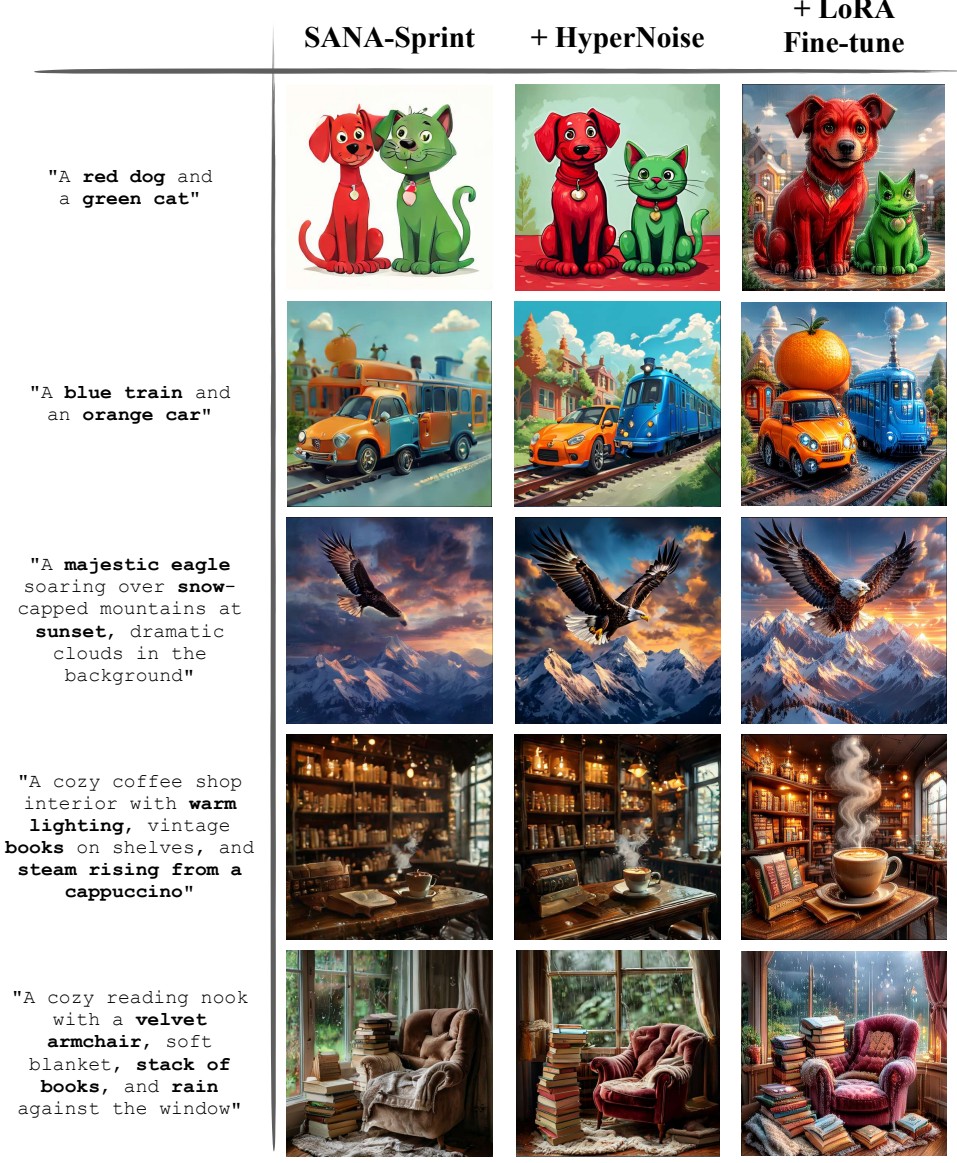

"A **red dog** and a **green cat**"

"A **blue train** and an **orange car**"

"A **majestic eagle** soaring over **snow**-capped mountains at **sunset**, dramatic clouds in the background"

"A cozy coffee shop interior with **warm lighting**, vintage **books** on shelves, and **steam rising from a cappuccino**"

"A cozy reading nook with a **velvet armchair**, soft blanket, **stack of books**, and **rain** against the window"

Figure 6: Examples of artifacts introduced by directly Direct Fine-tuning diffusion models on rewards [14, 53, 74] for the same reward objective in comparison to Noise Hypernetwork training with same initial noise.

## C.2 Diversity Analysis

We also investigate the impact of the diversity of the generated outputs as the result of our hypernetwork. For this purpose, we generate 50 images by varying the seed from the 553 prompts of the GenEval benchmark. The average similarity of different images for the same prompt are measured using similarities from

Table 7: We measure the average LPIPS and DINO similarity scores over images generated for 50 different seeds for the 553 prompts from GenEval.

| | LPIPS ↑ | DINO ↓ |
|---|---|---|
| SANA-Sprint | 0.608 ±0.074 | 0.780 ±0.103 |
| **+ Noise HyperNetwork** | 0.592 ±0.059 | 0.825 ±0.090 |

LPIPS [106] and DINOv2 [69] embeddings. The results in Tab. 7 indicate that the noise hypernetwork does not cause any collapse due to "reward-hacking" and broadly, the diversity of the generated images is in the same ballpark as the base model.

## C.3 Multi-step analysis

Here, in addition to the main text Table 2, we analyze the behavior of Noise Hypernetworks when moving beyond the few-step regime of $1 - 4$ steps. Remarkably, even when going up to 32 inference steps, we find that Noise Hypernetworks trained with the one-step generator, improve performance. We find that as we increase the NFEs, the added performance boost of the Noise Hypernetwork reduces. However, note that the underlying model SANA-sprint [13] was not trained to be used in the multi-step regime, but specifically for few-step generation.

## C.4 Challenges with Direct Fine-tuning

We also qualitatively illustrate the problems with directly fine-tuning diffusion models on differentiable rewards in Figure 6. As visualized, there are drastic artifacts introduced on the image which significantly contribute to improving the reward

Table 8: Mean GenEval results for SANA-Sprint highlighting generalization across inference timesteps of our Noise Hypernetwork.

| SANA-Sprint [13] | NFEs | GenEval Mean↑ |
|---|---|---|
| One-step | 1 | 0.70 |
| + Direct fine-tune [74] | 1 | 0.67 |
| **+ Noise Hypernetwork** | **2** | **0.75** |
| Two-step | 2 | 0.72 |
| + Direct fine-tune [74] | 2 | 0.66 |
| **+ Noise Hypernetwork** | **3** | **0.76** |
| Four-step | 4 | 0.73 |
| + Direct fine-tune [74] | 4 | 0.62 |
| **+ Noise Hypernetwork** | **5** | **0.77** |
| Eight-step | 8 | 0.74 |
| **+ Noise Hypernetwork** | **9** | **0.76** |
| Sixteen-step | 16 | 0.73 |
| **+ Noise Hypernetwork** | **17** | **0.75** |
| Thirty-two-step | 32 | 0.71 |
| **+ Noise Hypernetwork** | **33** | **0.72** |

scores. These artifacts are very similar to the ones noticed in several works [14, 41, 53] and require the development of several regularization strategies to address these issues. However as explained in Section 2, in the few-step regime the KL regularization term to the base model is difficult to be made tractable and thus, to the best of our knowledge there exists no theoretical grounded approach to learn the reward tilted distribution (Equation 3) with a one-step generator. The Noise Hypernework strategy on the other hand, ensures that the images remain in the original data distribution with its principled regularization.

## C.5 LoRA Rank analysis

Here, we ablate the LoRA rank for both HyperNoise and direct fine-tuning on SANA-Sprint. We find that a rank of 64 also seems to be sufficient to achieve almost the same improvements as rank 128, while a lower rank seems not to be expressive enough. On the other hand, fine-tuning seems to be suffering from increased overfitting on the reward.

## C.6 Qualitative Results

We provide additional qualitative samples for the base SANA-Sprint result along with the generation with our proposed noise hypernetwork in Figures 6 and 8. We broadly observe improved prompt following as well as superior visual quality in the generated images.

Table 9: GenEval results for HyperNoise on SANA-Sprint, showing generalization across timesteps.

| Method | NFEs | GenEval Mean↑ |
|---|---|---|
| SANA-Sprint (One-step) | 1 | 0.70 |
| LoRA-Rank 128 + HyperNoise | 2 | 0.75 |
| LoRA-Rank 64 + HyperNoise | 2 | 0.75 |
| LoRA-Rank 16 + HyperNoise | 2 | 0.71 |
| LoRA-Rank 8 + HyperNoise | 2 | 0.70 |
| SANA-Sprint (Two-step) | 2 | 0.72 |
| HyperNoise | 3 | 0.76 |
| SANA-Sprint (Four-step) | 4 | 0.73 |
| HyperNoise | 5 | 0.77 |
| HyperNoise (LoRA-Rank=64) | 5 | 0.76 |

Table 10: GenEval results for direct LoRA fine-tuning on SANA-Sprint.

| Method | NFEs | GenEval Mean↑ |
|---|---|---|
| SANA-Sprint (One-step) | 1 | 0.70 |
| LoRA-Rank 128 + LoRA fine-tune | 1 | 0.67 |
| LoRA-Rank 64 + LoRA fine-tune | 1 | 0.68 |
| LoRA-Rank 16 + LoRA fine-tune | 1 | 0.65 |
| LoRA-Rank 8 + LoRA fine-tune | 1 | 0.59 |
| SANA-Sprint (Two-step) | 2 | 0.72 |
| LoRA fine-tune | 2 | 0.66 |
| SANA-Sprint (Four-step) | 4 | 0.73 |
| LoRA fine-tune | 4 | 0.62 |

SANA-Sprint  + Noise Hypernetwork

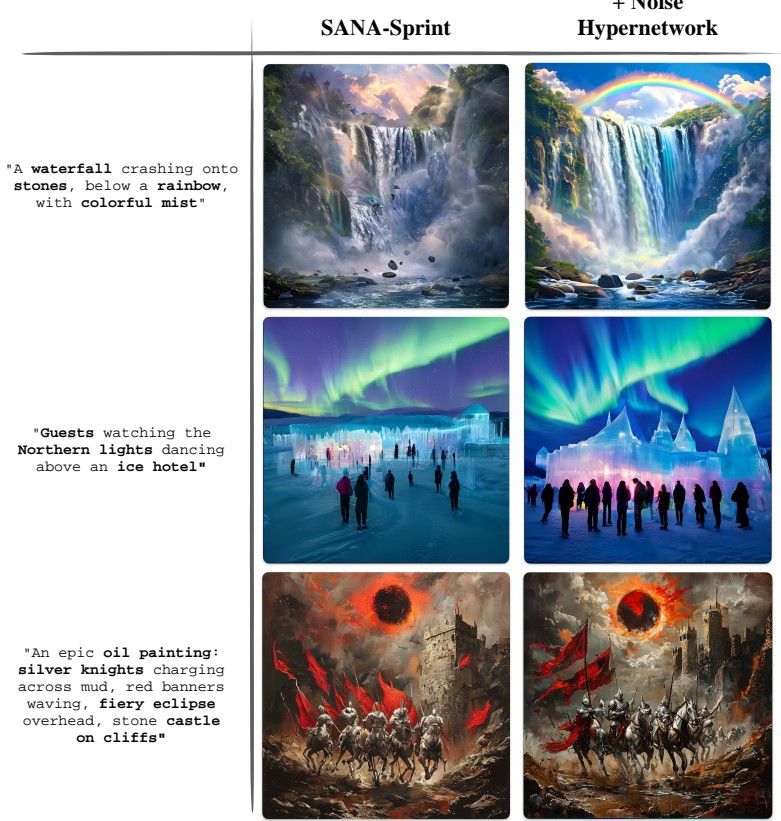

"A **waterfall** crashing onto **stones**, below a **rainbow**, with **colorful mist**"

"**Guests** watching the **Northern lights** dancing above an **ice hotel**"

"An epic **oil painting**: **silver knights** charging across mud, red banners waving, **fiery eclipse** overhead, stone **castle on cliffs**"

Figure 7: More qualitative results on the human-preference reward setting. Base SANA-Sprint compared to HyperNoise with same initial noise.

**SANA-Sprint**    **SANA-Sprint + HyperNoise**

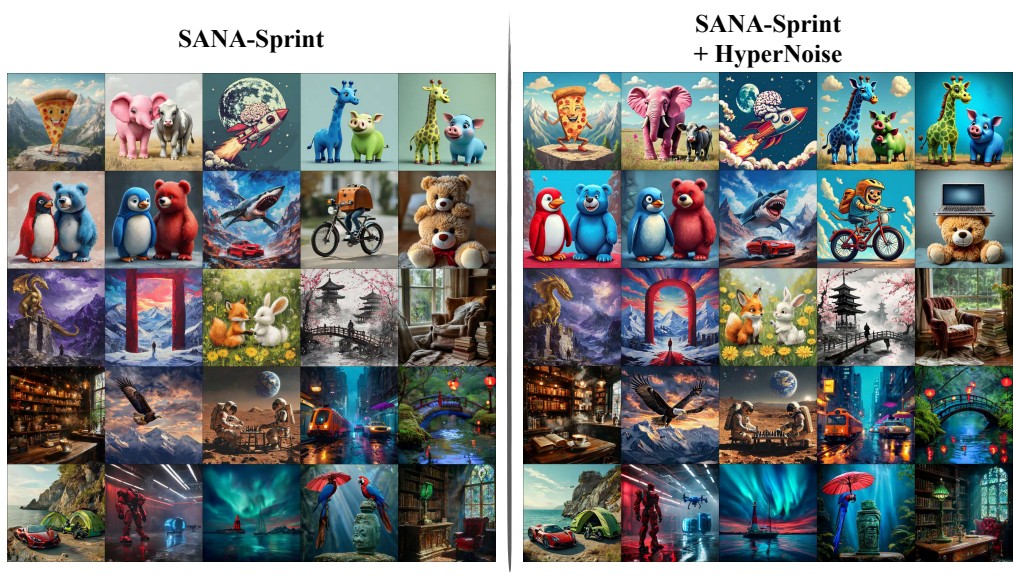

Figure 8: Non-cherry picked results on the human-preference reward setting. Base SANA-Sprint compared to HyperNoise with same initial noise.

