# OpenReview forum: "Noise Hypernetworks: Amortizing Test-Time Compute in Diffusion Models"
_NeurIPS.cc/2025/Conference — NeurIPS 2025 poster_

### Official Review · Reviewer_sVrs · 2025-06-26

**Clarity:** 2
**Significance:** 2
**Originality:** 2
**Rating:** 5
**Confidence:** 2

**Summary:**

The paper introduces a new method called Noise Hypernetworks, aiming to incorporate the advantages of test-time scaling into the training phase of diffusion models. Noise Hypernetworks learns a new distribution for the initial noise, producing modulated initial noise through a residual formulation. The loss function is based on theoretical results that aim to minimize the KL divergence between the initial noises while simultaneously maximizing the reward for the diffusion outputs. Experimental results demonstrate that Noise Hypernetworks can generate high-reward samples without compromising overall image quality.

**Questions:**

See weaknesses.

**Ethical Concerns:**

["NO or VERY MINOR ethics concerns only"]

**Final Justification:**

The majority of my concerns have been addressed through this rebuttal and responses to other reviewers, so I have decided to raise the score from 4 to 5.

**Limitations:**

The authors do discuss the limitations related to the need for improved reward models.

**Quality:**

2

**Strengths And Weaknesses:**

### Strengths

The paper is well written and easy to follow. The authors offer comprehensive theoretical justifications for their approach to tuning the initial noise distribution, establishing a solid theoretical foundation. The illustrative experiments are interesting, providing an intuitive insight into the method's performance.

### Weaknesses

1. Since the reward model is applied to the output samples of diffusion models, training the Noise Hypernetwork still requires costly backpropagation through the sampling process. This may restrict the training of the Noise Hypernetwork to one-step generation models, as indicated in Table 2. What would happen if the Noise Hypernetwork were trained on multi-step generation process? Would it still achieve high performance?

2. The performance may heavily depend on the quality of the distilled generative model $g_\theta$.

3. Is it unclear that whether altering the initial noise distribution could decrease the diversity of the generated samples.

---

> ### Author Rebuttal · Authors · 2025-07-31
>
> We thank the reviewer for their thoughtful assessment and for recognizing that our work is "well written" with "comprehensive theoretical justifications" and "interesting illustrative experiments." We appreciate their constructive feedback and address their specific concerns below.
>
> >***"... What would happen if the Noise Hypernetwork were trained on multi-step generation process? Would it still achieve high performance?"***
>
> The reviewer raises an excellent question about multi-step training feasibility. While multi-step training is theoretically possible through direct multi-step backpropagation or techniques like Adjoint Matching [1], which can compute gradients through multi-step ODE/SDE solvers efficiently, it remains unclear whether direct fine-tuning approaches or our hypernetwork method would be more effective in such regimes. While Adjoint Matching can directly learn reward-tilted distributions for multi-step generators through tractable KL regularization, their SDE-based formulation requires continuous-time dynamics making it computationally expensive and inapplicable to step-distilled models.
>
> - **Addressing a Critical Gap for Distilled Models**: Our work specifically targets distilled models, which represent an increasingly important class of generators due to their efficiency. This addresses a fundamental limitation where existing reward optimization methods require multi-step sampling processes that don't exist in distilled models.
>
> - **Practical Advantages of Our Approach**: Beyond theoretical considerations, our method offers compelling practical benefits. While our training is performed on one-step distilled generators, **it generalizes remarkably well to multi-step inference** as demonstrated in Tables 2 & 8, achieving significant improvements when deployed with 2-16 sampling steps. This approach maximizes training efficiency while maintaining deployment flexibility across different quality-speed trade-offs.
>
> - **Computational Efficiency**: Our approach also offers computational advantages during training. Rather than requiring expensive multi-step backpropagation through sampling processes, we can train efficiently on distilled models and still achieve strong performance across different inference regimes. This makes reward optimization accessible for the fast, practical generators that are increasingly dominant in real-world applications.
>
> ---
>
> > ***"The performance may heavily depend on the quality of the distilled generative model"***
>
> The reviewer correctly identifies that our method's performance depends on the quality of the underlying base model. We acknowledge this fundamental limitation of any post-training alignment approach. However, our experiments demonstrate consistent relative improvements across different base model architectures and scales, suggesting that our approach successfully identifies and exploits suboptimal regions in the learned noise-to-image mapping regardless of the base model's absolute capability level.
>
> ---
>
> > ***"Is it unclear that whether altering the initial noise distribution could decrease the diversity of the generated samples."***
>
> We thank the reviewer for this important consideration about diversity preservation. As detailed in Appendix C.2 (Table 7), we conduct comprehensive diversity analysis using LPIPS and DINO similarity metrics across 500+ prompts with 50 images per prompt. Our results show a nuanced diversity profile: while DINO similarity shows a modest decrease, LPIPS actually indicates a slight increase. Crucially, there is **no diversity collapse**. This controlled diversity trade-off is accompanied by meaningful quality improvements, representing principled optimization toward higher-reward regions rather than problematic mode collapse. This behavior is attributable to our KL regularization term that prevents the learned noise distribution from deviating too far from the original Gaussian prior.
>
> [1] Domingo-Enrich et al. "Adjoint Matching: Fine-tuning Flow and Diffusion Generative Models with Memoryless Stochastic Optimal Control". NeurIPS 2024.

---

### Official Review · Reviewer_sWqF · 2025-07-02

**Clarity:** 3
**Significance:** 3
**Originality:** 3
**Rating:** 4
**Confidence:** 4

**Summary:**

The authors solve the critical problem of integrating test-time scaling knowledge into a model during post-training. The authors replace reward guided test-time noise optimization in diffusion models with a Noise Hypernetwork that modulates initial input noise. It demonstrates through extensive experiments significant enhancements in generation quality for state-of-the-art distilled models

**Questions:**

Refer to weakness

**Ethical Concerns:**

["NO or VERY MINOR ethics concerns only"]

**Limitations:**

Yes

**Paper Formatting Concerns:**

/

**Quality:**

3

**Strengths And Weaknesses:**

strength
1. The authors predict an optimized initial noise for a fixed distilled generator with Lora, and theoretically learn through a tractable noise-space objective
2. The authors benchmark the noise hypernetwork against established methods, primarily direct LoRA fine-tuning of the base generative model and demonstrate the effectiveness of the proposed noise network.


weakness
1. How do you avoid the risk of significant deviation from the original data distribution? I saw the ImageReward figure in the experiment section, can the authors elaborate more on it?
2. Did the authors have the ablations on Hypernetworks's architecture? The authors now use Lora, and how does the number of rank or learnable prompts affect the results?

---

> ### Author Rebuttal · Authors · 2025-07-31
>
> We sincerely thank the reviewer for their positive assessment and for their constructive technical questions. We are glad they recognized that our work "solves the critical problem of integrating test-time scaling knowledge into a model during post-training" with "significant enhancements in generation quality."
>
> We address their specific concerns below, including results from a new ablation study prompted directly by their valuable feedback.
>
> ---
>
> >"***How do you avoid the risk of significant deviation from the original data distribution? I saw the ImageReward figure in the experiment section, can the authors elaborate more on it?"***
>
> We thank the reviewer for this critical question about maintaining fidelity to the original data distribution. This is indeed a fundamental challenge when optimizing generative models with external rewards, and our framework addresses it through principled regularization mechanisms.
>
> • **The Core Challenge.** When fine-tuning generative models to optimize specific rewards, there is an inherent risk that the model will find solutions outside the natural data distribution by achieving high reward scores through unrealistic or artifact-heavy samples that deviate significantly from the training manifold. This distributional drift can lead to outputs that, while scoring well on the target metric, lack the quality and realism of the original model.
>
> • **Our Solution: Noise-Space KL Regularization.** We address this challenge through explicit regularization in our objective function (Equation 7). The KL divergence term $D_{KL}(p_0^{\phi} \|\| p_0)$ directly constrains the learned noise distribution to remain close to the original Gaussian prior. By preventing the optimized noise from deviating too far from the initial noise, we mathematically constrain the model's outputs $g_\theta(\hat{x}_0)$ to remain close to the base data manifold.
>
> • **Quantitative and Qualitative Validation:**  The "redness" experiment provides compelling empirical proof of this mechanism. As the reviewer noted in **Figure 4**, direct fine-tuning (which lacks our regularization) leads to a catastrophic drop in the general ImageReward score as it over-optimizes for redness. This is visually confirmed in **Figure 3**, where direct fine-tuning deteriorates into unrealistic, monochromatic images, demonstrating a complete departure from the learned image manifold. In stark contrast, our method achieves the target reward while maintaining a high and stable ImageReward score, with generated images that preserve natural texture and semantic structure. This confirms our approach successfully optimizes _within_ the learned data distribution, rather than deviating from it. This principle of controlled optimization extends to our main results, as evidenced by the consistent quality improvements on the GenEval benchmark.
>
> • **Theoretical Guarantees.** This approach is grounded in the Data Processing Inequality (Equation 12), which provides a formal guarantee: minimizing KL divergence in noise space upper-bounds the KL divergence in the final image space. Crucially, unlike data-space KL terms that are intractable for distilled models, our noise-space formulation can be efficiently computed as an L2 penalty (Equation 10).
>
> • **Broader Validation.** This controlled optimization behavior extends beyond the redness experiment. Our GenEval results across multiple evaluation dimensions similarly show consistent quality improvements without the artifacts or unrealistic outputs typically associated with distributional deviation in reward-optimized models as visualized in Figure 6 (in the Appendix).
>
> ---
>
> > ***LoRA Architecture Choice and Rank Ablation***
>
> We thank the reviewer for this excellent question on our architectural choices. The selection of a LoRA-based architecture is a deliberate and critical decision, driven by the fundamental need to produce **prompt-dependent** noise modifications that can generalize to unseen prompts.
>
> **Prompt-Dependent Noise and Generalization**:
> - The central task of our Noise Hypernetwork, $f_\phi$, is to predict an optimal noise perturbation $f_\phi(x_0|c)$ that is conditioned on the semantic content of a given text prompt, $c$. The optimal noise required to improve an image of "a pink elephant and a grey cow" is fundamentally different from the noise needed to enhance "a brain riding a rocketship towards the moon."
>
> - Training a standalone network to learn this complex mapping from scratch would be exceptionally difficult. To achieve robust generalization to the vast space of unseen, open-world prompts, such a network would require a prohibitively large and diverse training dataset of prompts and a corresponding long training time.
>
> **LoRA as the Principled Solution**:
> - Our choice of a LoRA-based architecture directly solves this challenge. Instead of attempting to learn text-conditioning from scratch, LoRA allows our Noise Hypernetwork to inherit and leverage the powerful, pre-existing text-conditioning pathways of the frozen base model. By adapting key components like the cross-attention layers, our network efficiently taps into the rich semantic understanding that the base model has already acquired.
>
> - This approach ensures that our predicted noise modifications are semantically aware and generalize effectively to novel prompts, without the need for an large-scale amount of training data. It is the most parameter-efficient and direct way to make the noise prediction $f_\phi(x_0|c)$  truly prompt-dependent. During our initial development, we explored different text-conditioned architectures trained from scratch but found these struggled to learn any meaningful text conditioning on small-scale prompt sets.
>
> ### **LoRA Rank Ablation**
>
> To directly answer the reviewer's question on architectural hyperparameters, we conducted a new ablation study on the LoRA rank using the SANA-Sprint model.
>
> | Method                             | NFEs | GenEval Mean↑ |
> | ---------------------------------- | ---- | ------------- |
> | SANA-Sprint Baseline               | 1    | 0.70          |
> | Noise Hypernetwork (LoRA-Rank 128) | 2    | **0.75**      |
> | Noise Hypernetwork (LoRA-Rank 64)  | 2    | **0.75**      |
> | Noise Hypernetwork (LoRA-Rank 16)  | 2    | 0.71          |
> | Noise Hypernetwork (LoRA-Rank 8)   | 2    | 0.70          |
>
> **Analysis of Ablation Results:** The new ablation results validate that while our chosen rank of 128 offers robust, high-quality performance, a rank of 64 achieves identical results (0.75 GenEval score). This is a valuable finding, prompted by the reviewer's question, that demonstrates the potential for greater parameter efficiency. Meanwhile, performance degrades at lower ranks (8-16), which likely lack the capacity to model the complex prompt-conditional noise modifications required. We thank the reviewer for pushing us to explore this dimension. We will add this full analysis to the Appendix.

---

> > ### Author Response · Authors · 2025-08-08
> >
> > Dear Reviewer sWqF,
> >
> > Thank you for your thoughtful review and constructive questions. We hope our rebuttal was able to address your concerns about distribution drift and LoRA architecture choices. We're particularly grateful that your feedback prompted us to conduct an additional LoRA rank ablation study, which yielded valuable insights.
> >
> > Please let us know if there are any remaining questions or if any aspects of our response need further clarification. We'd be happy to provide additional details.
> >
> > Best regards,
> > The Authors

---

### Official Review · Reviewer_YHFj · 2025-07-03

**Clarity:** 4
**Significance:** 4
**Originality:** 3
**Rating:** 6
**Confidence:** 4

**Summary:**

We would often like to tweak a trained diffusion model so that it is more likely to produce certain kinds of outputs; for example, we might prefer if its outputs are somewhat more 'red' than usual. While this can in principle be done via prompting, the current excitement over test-time scaling has suggested it might also be done by spending more compute at inference time. One idea is to, at inference time, push a diffusion model towards certain kinds of samples by training the noise seed to reflect a reward model. While this works, solving an optimization problem like this at inference time can be somewhat time-consuming.

The authors propose to amortize this computation by training a "noise hypernetwork", i.e., to learn a reward-sensitive noise seed through additional training time. Importantly, this involves training an additional function, but not changing the weights of the trained diffusion model. They show that this approach works well in practice and has a reasonable theoretical basis.

**Questions:**

Can the authors link their core idea to work on the 'priors' of diffusion models? It seems like changing the prior (to reflect the reward function of interest) is precisely what the method does, since it doesn't actually change the downstream sampling process at all. The idea made me think of the (now kind of 'old') PriorGrad paper, although there are some key technical differences, like the fact that in that paper the changed prior also changed the training process.

**Ethical Concerns:**

["NO or VERY MINOR ethics concerns only"]

**Final Justification:**

The authors addressed the (admittedly minor) concerns I brought up related to novelty/motivation, the relationship between this work and diffusion model priors, and the flexibility vs efficiency tradeoff. I still feel like the paper is well-written and that the authors' proposed method has a solid quantitative performance.

**Limitations:**

This is obvious, but it may be worth mentioning that the proposed approach reduces flexibility in order to improve efficiency. If for some reason the user wants to use a variety of reward functions, one has to train a model for each. The inference time approach is much more costly, but (as a result) doesn't have this issue.

**Paper Formatting Concerns:**

No concerns.

**Quality:**

4

**Strengths And Weaknesses:**

**Strengths.**
- The paper is clear and well-written. The idea makes a lot of sense and has an appealing link to modifying the 'prior' of diffusion models, as well as previous work on spending test-time compute.
- The authors do a lot of work to test that their idea performs well, and provide both numbers and qualitative examples to show that it does. I was pretty impressed by some of these demonstrations (e.g., Fig 1, Figs 3 through 5).
- To validate their method, they do experiments to check for 'dumb' possibilities. For example, maybe noise hypernetworks help, but only about as much as direct fine-tuning. Table 2 shows that this isn't true. Table 1 makes a variety of important points that make sense; for example, spending lots of inference time compute a la ReNO still performs better than the noise hypernetwork approach, although not by an extreme amount. All of these experiments together paint a convincing picture that the method works well, even compared to many existing alternatives.
- Some of the precise numerical claims are impressive, e.g., the NFEs statement on line 281.


**Weaknesses.** I don't have much to say here. One thing that kind of bugged me was the presented motivation for this approach. The authors discuss excitement over improving performance by increasing inference time compute, which works but has the obvious drawback that inference time and cost go up. Then they say that they can mitigate this issue by spending that compute during training instead. But this logic is weird to me, since getting a model to do things efficiently by spending a lot of resources during training has been business as usual for a long time! It might be better to argue that, given how efficient the standard approach is, it's worth seeing if it can also be used to make the noise seed reward-sensitive.

The paper may contain some small typos. One I noticed is an apostrophe issue on line 336.

---

> ### Author Rebuttal · Authors · 2025-07-31
>
> We sincerely thank the reviewer for their thoughtful and encouraging feedback. We're delighted that the reviewer found our paper "clear and well-written" with an idea that "makes a lot of sense" and demonstrations that were "pretty impressive." We appreciate the detailed evaluation and address the points raised below.
>
> **On Motivation and Framing:**
>
> We thank the reviewer for this valuable perspective on our motivation. The reviewer is absolutely right that spending training resources to improve efficiency isn't new. What makes our approach distinctive is that we're specifically translating insights from successful test-time scaling methods into a post-training framework. Rather than designing optimization strategies from scratch, we're taking proven test-time techniques (like ReNO) and "compiling" them into efficient networks that preserve their benefits without the inference cost.
>
> This represents a fundamentally different paradigm from standard training: we're adapting and amortizing successful test-time innovations rather than traditional end-to-end optimization. The key insight is that test-time scaling successes can be translated into post-training adaptations that maintain the core benefits while dramatically improving practical deployment.
>
> **On Connection to Diffusion Priors:**
>
> We appreciate this insightful connection to diffusion priors. Our method does fundamentally modify the effective prior distribution of the diffusion process without requiring changes to the base model training or sampling procedure. This connects our work to prior methods that learn adaptive prior distributions for diffusion models. Our approach achieves prior modification through post-training adaptation while this line of work typically modifies the training process itself or requires architectural changes to the base model, e.g. [1, 2]. We'll incorporate this connection to adaptive prior methods in our related work section to better position our contribution within this line of work.
>
> _**On Flexibility vs. Efficiency Trade-off:**_
>
> The reviewer raises an important consideration about our approach requiring separate models for different reward functions, which trades flexibility for efficiency compared to test-time methods. This trade-off becomes highly favorable for production scenarios with stable reward functions. We discuss this in detail in our response to Reviewer RnrZ, including a comprehensive cost-benefit analysis, which we will add to the appendix. In summary, that analysis shows:
> - **Scope:** Our method is ideal for production scenarios with a well-defined, valuable reward function that needs to be deployed at scale (e.g., human-preference models, brand styles, or photorealism enhancers).
> - **Amortization:** For complex rewards, the one-time training cost is fully recovered after generating approximately 50k images, leading to very significant long-term savings in both cost and latency.
>
> Lastly, we thank the reviewer for catching the apostrophe issue on line 336 and will fix this along with a careful proofreading pass.
>
> [1] Lee et al. "PriorGrad: Improving Conditional Denoising Diffusion Models with Data-Dependent Adaptive Prior". ICLR 2022.
>
> [2] Bartosh et al. "Neural Flow Diffusion Models: Learnable Forward Process for Improved Diffusion Modelling". NeurIPS 2024.

---

> > ### Comment · Reviewer_YHFj · 2025-08-07
> >
> > Thanks to the authors for their thoughtful response. I see now that my perspective on the relationship between the authors' work and test-time vs training compute was a bit wrong, and now understand better some of the novelty of this work.
> >
> > I'm happy with the treatment of the flexibility vs efficiency tradeoff, I think this makes for a good addition to the paper.
> >
> > Overall, I still think the paper is nice and that it should definitely be accepted.

---

### Official Review · Reviewer_RnrZ · 2025-07-12

**Clarity:** 3
**Significance:** 3
**Originality:** 3
**Rating:** 4
**Confidence:** 4

**Summary:**

This paper addresses the problem of test-time scaling for distilled diffusion models, with the goal of quickly adapting pretrained image generators to align with user-specific preferences. Instead of relying on direct fine-tuning of the base model or performing costly per-sample test-time noise optimization, the authors propose a “pre-trained” LoRA-based Noise Hypernetwork that learns to modulate the input noise in accordance with a given reward function. By keeping the base model parameters fixed and learning only a lightweight noise transformation network, the method significantly reduces inference-time overhead while preserving or improving output quality. The key contribution lies in a theoretically grounded training objective that includes a tractable KL regularization in the noise space to ensure fidelity to the original model's distribution.

**Questions:**

1. The method requires training a separate hypernetwork for each reward function. While this is understandable, it limits generality—especially since simple reward functions (e.g., color biases) are not meaningful, and complex ones (e.g., composed human preference models like those in ReNO) tend to yield narrowly specialized models with limited generalization. It would be helpful if the authors could discuss the intended scope of applicability and limitations of the proposed framework.
2. While ReNO requires expensive test-time optimization, it is end-to-end and can be applied on a per-sample basis. The proposed method, by contrast, incurs a one-time training cost. Could the authors elaborate on the relative advantages in low-resource vs. large-scale generation settings, and provide a cost-benefit comparison (e.g., amortized training time vs. repeated test-time inference)?
3. In Figure 3, although the method preserves the global image structure better than LoRA fine-tuning, it appears that the main object in the image becomes progressively smaller during training. Does this suggest that the reward function (e.g., “redness”) is insufficiently informative, encouraging the model to optimize a narrow visual cue rather than a meaningful preference? This seems to suggest a form of reward hacking; could the authors comment on how robust the method is under poorly specified or degenerate rewards?

**Ethical Concerns:**

["NO or VERY MINOR ethics concerns only"]

**Limitations:**

yes

**Quality:**

3

**Strengths And Weaknesses:**

Strengths:
1. The theoretical motivation and derivation of the method are solid and rigorous, particularly the approximation of the KL divergence in the noise space and the use of the data processing inequality for distributional control.
2. The core idea—to amortize test-time noise optimization via a pretrained hypernetwork—is novel, elegant, and highly practical.
3. The KL divergence regularization is applied in a clever and tractable form, leading to better preservation of image quality while optimizing for reward-aligned outputs.
4. The paper is well-written and well-structured, making the contributions easy to understand and follow.

Weaknesses:
1. The main comparison is against ReNO. While this is a reasonable choice given the shared setup, the paper could benefit from broader qualitative comparisons, e.g., prompting-based baselines or naive fine-tuning.
2. The paper emphasizes inference-time efficiency, but provides limited discussion of the cost and scalability of training the noise hypernetwork itself, especially with large or complex reward models.

---

> ### Author Rebuttal · Authors · 2025-07-31
>
> We sincerely thank the reviewer for their thorough and constructive feedback. We're pleased that the reviewer found our theoretical contributions "solid and rigorous" and our core idea "novel, elegant, and highly practical." Below, we address all the points raised.
>
> > "... ***the paper could benefit from broader qualitative comparisons, e.g., prompting-based baselines or naive fine-tuning.***
>
> We thank the reviewer for this valuable suggestion. To clarify, our method is the first to learn reward-optimized distributions for distilled diffusion models, which makes defining direct baselines challenging. However, we agree that a broader comparison against different test-time methods is beneficial here. Thus, we have now benchmarked our approach against two additional test-time techniques in the human-preference reward setting:
> 1. **LLM-based prompt optimization** following [1, 2]. The prompt optimization is handled by an LLM (Llama 3.1 8B), which is tasked to optimize a score, where we employ the same reward ensemble as used for ReNO and Noise Hypernetworks. For a fair comparison, we benchmark it with a total of 50 evaluated images (same as for ReNO), where in one iteration, the LLM proposes 5 new prompts, and we run for 10 iterations. We used the parameters from the official MILS repository [2], modifying only the reward ensemble and number of generated images per prompt.
> 2. **Best-of-N sampling** as e.g. used in [3]. Here we sample N=50 noises for each prompt and choose the best image based on, again, the used reward ensemble.
> #### Results on SANA-Sprint
>
> | Method                           | Inference Time (s) | Relative Speed (vs. Base) | GenEval Score |
> | :------------------------------- | :----------------: | :-----------------------: | :-----------: |
> | SANA-Sprint                      |        0.2         |          **1x**           |     0.70      |
> | **+ Noise Hypernetworks (Ours)** |      **0.3**       |         **1.5x**          |   **0.75**    |
> | *+ Best-of-N*                    |       ~15.0        |           ~75x            |     0.79      |
> | *+ Prompt Optimization*          |       ~115.0       |           ~575x           |     0.75      |
> | *+ ReNO*                         |       ~30.0        |           ~150x           |     0.81      |
>
> #### Results on SD-Turbo
>
> | Method                           | Inference Time (s) | Relative Speed (vs. Base) | GenEval Score |
> | :------------------------------- | :----------------: | :-----------------------: | :-----------: |
> | SD-Turbo                         |        0.2         |          **1x**           |     0.49      |
> | **+ Noise Hypernetworks (Ours)** |      **0.3**       |         **1.5x**          |   **0.57**    |
> | *+ Best-of-N*                    |       ~10.0        |           ~50x            |     0.60      |
> | *+ Prompt Optimization*          |       ~110.0       |           ~550x           |     0.59      |
> | *+ ReNO*                         |       ~20.0        |           ~100x           |     0.63      |
>
> - These results highlight the practical efficiency advantage of our method. Noise Hypernetworks achieves significant quality improvements with only a 1.5x latency increase, while alternative test-time methods require 50x to 575x more computation time. Notably, our approach achieves similar results to LLM-based prompt optimization while being ~380x faster, making it practical for real-time applications where other methods would be prohibitive.
>
> - Beyond direct comparison, our approach offers a unique advantage: it's fully compatible with existing test-time methods. Test-time approaches like ReNO could initialize from our network's predicted noise rather than random noise, potentially achieving better results or faster convergence by starting from a more promising region.
>
> **Fine-tuning comparison**: Finally, to demonstrate the importance of our method's formulation, we also benchmarked against direct LoRA fine-tuning (detailed in Section 4.1). This approach consistently fails to learn the reward-tilted distribution, as the absence of our proposed KL-regularization term leads to a divergence from the natural image manifold (Figure 3). We observe this same failure mode in the human-preference setting (Table 2), with visual examples of image degradation from reward-hacking provided in the Appendix (Figure 6). This underscores that simply fine-tuning with LoRA is insufficient and highlights the critical role of our proposed KL-regularization term in preventing reward-hacking and preserving image quality.
>
> ---
>
> _**On Training Cost, Generality, and Amortization:**_
>
> We thank the reviewer for raising these important points about our method's practical applicability. We'll address them together since they're closely related.
>
> - **Training Cost & Scalability:** First, some context on the computational requirements. Training our LoRA-based Noise Hypernetwork takes about 10 minutes on 1 H100 GPU for simple rewards (like redness) and 34 hours on 6 H100 GPUs for human preference rewards. While 34 hours sounds substantial, this is a one-time cost that's small compared to training the base diffusion models themselves. More importantly, this upfront investment "compiles" a complex iterative optimization process into a single efficient network that adds almost no latency at inference.
>
> - **Amortized Cost-Benefit:** The economics become compelling at scale. Our training uses 204 H100-GPU hours total. At inference, ReNO takes ~30 seconds per image on an A100 while inference with our Hypernetwork takes only ~0.3 seconds. Using a conservative 1:2 price ratio for A100 vs H100, our training investment equals 408 A100-equivalent hours. Since we save 29.7 seconds of A100 time per image, the training cost is fully recovered after generating about **50k images**. For any application generating hundreds of thousands of images, this represents massive long-term savings in both cost and latency.
>
> - **Scope and Generality:** This analysis highlights where our method shines: scenarios with a well-defined, valuable reward function that needs to be deployed at scale (e.g., human-preference reward models, a complex brand style, or a photorealism enhancer). While test-time optimization offers more flexibility for exploration, our approach is designed for the regime where a certain reward has been identified and now needs to be deployed efficiently at a larger scale.
>
> We'll add this detailed cost-benefit analysis and scope discussion to the appendix of our revised manuscript.
>
> ---
>
> > ***"... it appears that the main object in the image becomes progressively smaller during training. Does this suggest that the reward function (e.g., “redness”) is insufficiently informative, encouraging the model to optimize a narrow visual cue rather than a meaningful preference? This seems to suggest a form of reward hacking; could the authors comment on how robust the method is under poorly specified or degenerate rewards?"***
>
> We thank the reviewer for pointing out the object shrinking behavior. This is actually a feature, not a bug, and shows why our approach is more robust than alternatives.
>
> The "redness" reward we use here is intentionally narrow. A model trying to maximize redness could easily turn everything red and destroy the image completely. What's noteworthy is that our method doesn't do this. Instead, it finds a clever workaround: shrink the object so more red background shows through. This satisfies the reward while keeping the image intact.
>
> Compare this to direct LoRA fine-tuning, which breaks down and diverges from the natural image manifold and thus, from the base model (Figure 3). Our KL-regularized version finds a reasonable solution that games the reward in a harmless way. The ImageReward scores in Figure 4 confirm that image quality stays stable throughout training.
>
> This behavior is exactly what we want from a robust system. Real-world reward functions are often imperfect or incomplete. When that happens, we need methods that bend rather than break. Our KL regularization provides exactly this by keeping the model close to its original distribution, preventing the kind of catastrophic failures that make direct fine-tuning impractical.
>
> ---
>
> [1] Mañas et al. "Improving Text-to-Image Consistency via Automatic Prompt Optimization". TMLR 2024.
>
> [2] Ashutosh et al. "LLMs can see and hear without any training". ICML 2025.
>
> [3] Ma et al. "Inference-Time Scaling for Diffusion Models beyond Scaling Denoising Steps". CVPR 2025.

---

> > ### Comment · Reviewer_RnrZ · 2025-08-04
> >
> > Thank you for the thorough and thoughtful rebuttal. I particularly appreciate the authors' discussion on reward hacking. I agree that in real-world applications, reward functions are often imperfect, underspecified, or even unintuitive. In such scenarios, maintaining output stability and avoiding catastrophic failures becomes a crucial property, and the proposed KL-regularized formulation appears to provide a reasonable and robust safeguard.
> >
> > That said, while the added comparisons and cost analysis are helpful, the benefits of the proposed method become clearly significant only at the scale of tens or hundreds of thousands of generations. This reinforces my original concern that its practical efficiency advantage may be limited to very specific deployment settings.
> >
> > Overall, I find the authors’ responses well-argued and convincing on several fronts, but I will maintain my original score.

---

> > > ### Author Response · Authors · 2025-08-04
> > >
> > > Thank you for the thoughtful feedback and for engaging so constructively with our rebuttal. We especially appreciate your positive assessment of our discussion on reward hacking and your view that our KL-regularized formulation provides a robust safeguard.
> > >
> > > Regarding your point about deployment settings, we'd like to share our perspective on how diffusion models are typically used in practice. While individual users certainly customize models for personal use, the bulk of generations come from deployed models that are downloaded and used widely.
> > >
> > > Popular fine-tuned models on platforms like Huggingface or Civitai routinely see tens of thousands of downloads, leading to millions of images generated across their user base. Commercial applications follow similar patterns. For these widely used models, per-image optimization costs become prohibitive. Our method enables reward optimization to work effectively in precisely these high-volume scenarios.
> > >
> > > So while individual customization represents one important use case that test-time methods serve well, we see our approach as addressing what appears to be the predominant mode of diffusion model usage.

---

### Note · Authors · 2025-08-16

We sincerely thank all reviewers and the AC for their constructive thorough evaluation of our work on Noise Hypernetworks.

**Key Improvements Made:** Following reviewer feedback, we have strengthened our paper with:

- Comprehensive comparisons against additional test-time methods (Best-of-N sampling, LLM-based prompt optimization), demonstrating our method achieves comparable quality at 50-575× faster inference
- LoRA rank ablation study showing rank-64 achieves identical performance to rank-128, offering greater parameter efficiency
- Expanded discussion of our method's scope and cost-benefit analysis for deployment scenarios
- Clarified connections to diffusion priors and the flexibility-efficiency trade-off


**Core Contribution Validated:** We are pleased reviewers recognized our theoretical framework as "solid and rigorous" (RnrZ), our core idea as "novel, elegant, and highly practical" (RnrZ), and our experimental validation as "impressive" (YHFj). Reviewers consistently praised the paper's clarity: "well-written and well-structured, making the contributions easy to understand" (RnrZ), "clear and well-written" (YHFj), and "well written and easy to follow" (sVrs).

The unanimous positive assessment reflects both the theoretical rigor and practical impact of our contribution. The method opens new directions for efficient model alignment, with immediate applications in human preference optimization and broader implications for the growing paradigm of test-time scaling.

Thank you for your consideration.

---

### Decision · Program_Chairs · 2025-09-17

**Decision:**

Accept (poster)

**Comment:**

The paper proposes a hypernetwork to predict residual noise from the original noise in the diffusion model to optimise certain reward functions. Compared to previous noise optimisation methods, the main contribution is to train the hypernetwork to avoid test-time optimisation to improve inference efficiency. The authors provide a very good story for this in the context of test-time scaling, which is a currently important topic, and the proposed hypernetwork moves the computation from test-time to post-training. Other implementation techniques include using LoRA and step-distilled generators in the hypernetwork and KL divergence regulation during training.

Reviewers consistently provided positive comments regarding the method, motivation, and writing quality. Reviewers queried about more experiments, applicability to multi-step generator, diversity of the examples, etc., which are mostly addressed in the rebuttal. The main reason for acceptance is that there is no major weakness in all aspects, and the method is motivated in the context of the trending test-time scaling.

Reviewers' suggestions should be incorporated into the next version of this paper. In addition, it will also be beneficial to add discussions on the generalisation ability of the hypernetwork to different reward functions once trained, and also the robustness of the hypernetwork to reward hacking compared to other existing techniques.